# Time-Constrained Robust MDPs

**Adil Zouitine**[*,1,2]**, David Bertoin**[*,1,3,6]**, Pierre Clavier**[*,4,5]
**Matthieu Geist**[7], **Emmanuel Rachelson**[2,6]

[1]IRT Saint-Exupéry, [2]ISAE-SUPAERO, Université de Toulouse,[3]IMT, INSA Toulouse
[4]École Polytechnique, CMAP, [5]Inria Paris, HeKA
[6]ANITI, [7]Cohere
{adil.zouitine, david.bertoin}@irt-saintexupery.com,

pierre.clavier@polytechnique.edu

## Abstract

Robust reinforcement learning is essential for deploying reinforcement learning algorithms in real-world scenarios where environmental uncertainty predominates. Traditional robust reinforcement learning often depends on rectangularity assumptions, where adverse probability measures of outcome states are assumed to be independent across different states and actions. This assumption, rarely fulfilled in practice, leads to overly conservative policies. To address this problem, we introduce a new time-constrained robust MDP (TC-RMDP) formulation that considers multifactorial, correlated, and time-dependent disturbances, thus more accurately reflecting real-world dynamics. This formulation goes beyond the conventional rectangularity paradigm, offering new perspectives and expanding the analytical framework for robust RL. We propose three distinct algorithms, each using varying levels of environmental information, and evaluate them extensively on continuous control benchmarks. Our results demonstrate that these algorithms yield an efficient tradeoff between performance and robustness, outperforming traditional deep robust RL methods in time-constrained environments while preserving robustness in classical benchmarks. This study revisits the prevailing assumptions in robust RL and opens new avenues for developing more practical and realistic RL applications.

## 1  Introduction

Robust MDPs capture the problem of finding a control policy for a dynamical system whose transition kernel is only known to belong to a defined uncertainty set. The most common framework for analyzing and deriving algorithms for robust MDPs is that of $sa$-rectangularity [1, 2], where probability measures on outcome states are picked independently in different source states and actions (in formal notation, $\mathbb{P}(s'|s,a)$ and $\mathbb{P}(s'|\bar{s},\bar{a})$ are independent of each other). This provides an appreciable decoupling of worst transition kernel search across time steps and enables sound algorithms like robust value iteration (RVI). But policies obtained for such $sa$-rectangular MDPs are by nature very conservative [3, 4], as they enable drastic changes in environment properties from one time step to the next, and the algorithms derived from RVI tend to yield very conservative policies even when applied to non-$sa$-rectangular robust MDP problems.

In this paper, we depart from the rectangularity assumption and turn towards a family of robust MDPs whose transition kernels are parameterized by a vector $\psi$. This parameter vector couples together the outcome probabilities in different $(s,a)$ pairs, hence breaking the independence assumption that is problematic, especially in large dimension [3]. This enables accounting for the notion of transition

---

[*]These authors contributed equally to this work

38th Conference on Neural Information Processing Systems (NeurIPS 2024).

model consistency across states and actions: outcome probabilities are not picked independently anymore but are rather set across the state and action spaces by drawing a parameter vector. In turn, we examine algorithms for solving such parameter-based robust MDPs when the parameter is constrained to follow a bounded evolution throughout time steps. Our contributions are the following.

1. We introduce a formal definition for parametric robust MDPs and time-constrained robust MDPs, discuss their properties and derive a generic algorithmic framework (Sec. 2).

2. We propose three algorithmic variants for solving time-constrained MDPs, named vanilla `TC` , `Stacked-TC` and `Oracle-TC` (Sec. 4), which use different levels of information in the state space, and come with theoretical guaranties (Sec. 6).

3. These algorithms are extensively evaluated in MuJoCo [5] benchmarks, demonstrating they lead to non-conservative and robust policies (Sec. 5).

## 2 Problem statement

**(Robust) MDPs.** A Markov Decision Process (MDP) [6] is a model of a discrete-time, sequential decision making task. At each time step, from a state $s_t \in S$ of the MDP, an action $a_t \in A$ is taken and the state changes to $s_{t+1}$ according to a stationary Markov transition kernel $p(s_{t+1}|s_t, a_t)$, while concurrently receiving a reward $r(s_t, a_t)$. $S$ and $A$ are measurable sets and we write $\Delta_S$ and $\Delta_A$ the set of corresponding probability distributions. A stationary policy $\pi(\cdot|s)$ is a mapping from states to distributions over actions, prescribing which action should be taken in $s$. The value function $v_p^\pi$ of policy $\pi$ maps state $s$ to the expected discounted sum of rewards $\mathbb{E}_{p,\pi}[\sum_t \gamma^t r_t]$ when applying $\pi$ from $s$ for an infinite number of steps. An optimal policy for an MDP is one whose value function is maximal in any state. In a Robust MDP (RMDP) [1, 2], the transition kernel $p$ is not set exactly and can be picked in an adversarial manner at each time step, from an uncertainty set $\mathcal{P}$. Then, the pessimistic value function of a policy is $v_{\mathcal{P}}^\pi(s) = \min_{p \in \mathcal{P}} v_p^\pi(s)$. An optimal robust policy is one that has the largest possible pessimistic value function $v_{\mathcal{P}}^*$ in any state, hence yielding an adversarial $\max_\pi \min_p$ optimization problem. Robust Value Iteration (RVI) [1, 7] solves this problem by iteratively computing the one-step lookahead best pessimistic value:

$$v_{n+1}(s) = T_{\mathcal{P}}^* v_n(s) := \max_{\pi(s) \in \Delta_A} \min_{p \in \mathcal{P}} \mathbb{E}_{a \sim \pi(s)}[r(s, a) + \mathbb{E}_p[v_n(s')]].$$

The $T_{\mathcal{P}}^*$ operator is called the robust Bellman operator and the sequence of $v_n$ functions converges to the robust value function $v_{\mathcal{P}}^*$ as long as the adversarial transition kernel belongs to the simplex of $\Delta_S$.

**Zero-sum Markov Games.** Robust MDPs can be cast as zero-sum two-players Markov games [8, 9] where $B$ is the action set of the adversarial player. Writing $\bar{\pi} : S \times A \to \Delta_B$ the policy of this adversary, the robust MDP problem turns to $\max_\pi \min_{\bar{\pi}} v^{\pi,\bar{\pi}}$, where $v^{\pi,\bar{\pi}}(s)$ is the expected sum of discounted rewards obtained when playing $\pi$ (agent actions) against $\bar{\pi}$ (transition models) at each time step from $s$. This enables introducing the robust value iteration sequence of functions

$$v_{n+1}(s) := T^{**} v_n(s) := \max_{\pi(s) \in \Delta_A} \min_{\bar{\pi}(s,a) \in \Delta_S} (T^{\pi,\bar{\pi}} v_n)(s)$$

where $T^{\pi,\bar{\pi}} := \mathbb{E}_{a \sim \pi(s)}[r(s, a) + \gamma \mathbb{E}_{s' \sim \bar{\pi}(s,a)} v_n(s')]$ is a zero-sum Markov game operator. These operators are also $\gamma-$contractions and converge to their respective fixed point $v^{\pi,\bar{\pi}}$ and $v^{**} = v_{\mathcal{P}}^*$ [9]. This formulation will be useful to derive a practical algorithm in Section 4.

Often, this convergence is analyzed under the assumption of $sa$-rectangularity, stating that the uncertainty set $\mathcal{P}$ is a set product of independent subsets of $\Delta_S$ in each $s, a$ pair. Quoting [1], rectangularity is a sort of independence assumption and is a minimal requirement for most theoretical results to hold. Within robust value iteration, rectangularity enables picking $\bar{\pi}(s_t, a_t)$ completely independently of $\bar{\pi}(s_{t-1}, a_{t-1})$. To set ideas, let us consider the robust MDP of a pendulum, described by its mass and rod length. Varying this mass and rod length spans the uncertainty set of transition models. The rectangularity assumption induces that $\bar{\pi}(s_t, a_t)$ can pick a measure in $\Delta_S$ corresponding to a mass and a length that are completely independent from the ones picked in the previous time step. While this might be a good representation in some cases, in general it yields policies that are very conservative as they optimize for adversarial configurations which might not occur in practice.

We first step away from the rectangularity assumption and define a parametric robust MDP as an RMDP whose transition kernels are spanned by varying a parameter vector $\psi$ (typically the mass

and rod length in the previous example). Choosing such a vector couples together the probability measures on successor states from two distinct $(s, a)$ and $(s', a')$ pairs. The main current robust deep RL algorithms actually optimize policies for such parametric robust MDPs but still allow the parameter value at each time step to be picked independently of the previous time step.

**Parametric MDPs.** A parametric RMDP is given by the tuple $(S, A, \Psi, p_\psi, r)$ where the transition kernel $p_\psi(s, a) \in \Delta_S$ is parameterized by $\psi$, and $\Psi$ is the set of values $\psi$ can take, equipped with an appropriate metric. This yields the robust value iteration update :

$$v_{n+1}(s) = \max_{\pi(s) \in \Delta_A} \min_{\psi \in \Psi} (T_\psi^\pi v_n)(s) := \max_{\pi(s) \in \Delta_A} \min_{\psi \in \Psi} \mathbb{E}_{a \sim \pi(s)}[r(s, a) + \gamma \mathbb{E}_{s' \sim p_\psi(s, a)} v_n(s')].$$

A parametric RMDP remains a Markov game and the Bellman operator remains a contraction mapping as long as $p_\psi$ can reach only elements in the simplex of $\Delta_S$, where the adversary's action set is the set of parameters instead of a (possibly $sa$-rectangular) set of transition kernels.

**Time-constrained RMDPs (TC-RMDPs).** We introduce TC-RMDPs as the family of parametric RMDPs whose parameter's evolution is constrained to be Lipschitz with respect to time. More formally a TC-RMDP is given by the tuple $(S, A, \Psi, p_\psi, r, L)$, where $\|\psi_{t+1} - \psi_t\| \leq L$, that is the parameter change is bounded through time. In the previous pendulum example, this might represent the wear of the rod which might lose mass or stretch length. Similarly, and for a larger scale illustration, TC-RMDPs enable representing the possible evolutions of traffic conditions in a path planning problem through a busy town. Starting from an initial parameter value $\psi_{-1}$, the pessimistic value function of a policy $\pi$ is non-stationary, as $\psi_0$ is constrained to lay at most $L$-far away from $\psi_{-1}$, $\psi_1$ from $\psi_0$, and so on. Generally, this yields non-stationary value functions as the uncertainty set at each time step depends on the previous uncertainty parameter. To regain stationarity without changing the TC-RMDP definition, we first change the definition of the adversary's action set. The adversary picks its actions in the constant set $B = \mathcal{B}(0_\Psi, L)$, which is the ball of radius $L$ centered in the null element in $\Psi$. In turn, the state of the Markov game becomes the pair $s, \psi$ and the Markov game itself is given by the tuple $((S \times \Psi), A, B, p_\psi, r)$, where the Lipschitz constant $L$ is included in $B$. Thus, given an action $b_t \in B$ and a previous parameter value $\psi_{t-1}$, the parameter value at time $t$ is $\psi_t = \psi_{t-1} + b_t$. Then, we define the pessimistic value function of a policy as a function of both the state $s$ and parameter $\psi$:

$$v_B^\pi(s, \psi) := \min_{\substack{(b_t)_{t \in \mathbb{N}}, \\ b_t \in B}} \mathbb{E}\Big[\sum \gamma^t r_t | \psi_{-1} = \psi, s_0 = s, b_t \in B, \psi_t = \psi_{t-1} + b_t, a \sim \pi, s_t \sim p_{\psi_t}\Big],$$

$$v_B^*(s, \psi) = \max_{\pi(s, \psi) \in \Delta_A} v_B^\pi(s, \psi).$$

In turn, an optimal robust policy is a function of $s$ and $\psi$ and the TC robust Bellman operators are:

$$v_{n+1}(s, \psi) := T_B^* v_n(s, \psi) := \max_{\pi(s, \psi) \in \Delta_A} T_B^\pi v_n(s, \psi),$$

$$:= \max_{\pi(s, \psi) \in \Delta_A} \min_{b \in B} \mathbb{E}_{a \sim \pi(s)}[r(s, a) + \gamma \mathbb{E}_{s' \sim p_{\psi + b}(s, a)} v_n(s', \psi + b)].$$

This iteration scheme converges to a fixed point according to Th. 2.1.

**Theorem 2.1.** *The time-constrained (TC) Bellman operators $T_B^\pi$ and $T_B^*$ are contraction mappings. Thus the sequences $v_{n+1} = T_B^\pi v_n$ and $v_{n+1} = T_B^* v_n$, converge to their respective fixed points $v_B^\pi$ and $v_B^*$.*

Proof of Th. 2.1 can be found in Appendix B. We refer to this formulation as algorithm `Oracle-TC` (see Section 4 for implementation details) since an oracle makes the current parameter $\psi$ visible to the agent. Therefore, it is possible to derive optimal policies for TC-RMDPs by iterated application of this TC Bellman operator. These policies have the form $\pi(s, \psi)$. In the remainder of this paper, we extend state-of-the-art robust deep RL algorithms to the TC-RMDP framework. In particular, we compare their performance and robustness properties with respect to classical robust MDP formulations, we also discuss their relation with the $\pi(s)$ robust policies of classical robust MDPs.

If the agent is unable to observe the state variable $\psi$, it is not possible to guarantee the existence of a stationary optimal policy of the form $\pi(s)$. Similarly, there is no guarantee of convergence of value functions to a fixed point. Nonetheless, this scenario, in which access to the $\psi$ parameter is not available, is more realistic in practice. It turns the two-player Markov game into a partially observable

Markov game, where one can still apply the TC Bellman operator but without these guarantees of convergence. We call vanilla `TC` the repeated application of the TC Bellman operator in this partially observable case. Vanilla `TC` will be tested in practice, and some theoretical properties of the objective function will be derived using the Lipschitz properties (Sec 6).

## 3    Related works

Since our method is a non-rectangular, Deep Robust RL algorithm, (possibly non-stationary for `Stacked-TC` and `TC` ), we discuss the following related work.

**Non-stationary MDPs.**  First, non-stationarity has been studied in the Bandits setting in [10]. Then, for episodic, non-stationary MDPs [11, 12, 13] have explored and provided regret bounds for algorithms that use oracle access to the current reward and transition functions. More recently [14, 15] have facilitated oracle access by performing a count-based estimation of the reward and transition functions based on the recent history of interactions. Finally, for tabular MDPs, past data from a non-stationary MDP can be used to construct a full Bayesian model [16] or a maximum likelihood model [17] of the transition dynamics. We focus on the setting not restricted to tabular representations

**Non-rectangular RMDPs.**  While rectangularity in practice is very conservative, it can be demonstrated that, in an asymptotic sense, non-rectangular ellipsoidal uncertainty sets around the maximum likelihood estimator of the transition kernel constitute the smallest possible confidence sets for the ground truth transition kernel, as implied by classical Cramér-Rao bounds. This is in accordance with the findings presented in § 5 and Appendix A of [7]. More recently, [3] extends the rectangular assumptions using a factored uncertainty model, where all transition probabilities depend on a small number of underlying factors denoted $\boldsymbol{w}_1, \ldots, \boldsymbol{w}_r \in \mathbb{R}^{\mathbb{S}}$, such that each transition probability $P_{sa}$ for every $(s, a)$ is a linear (convex) combination of these $r$ factors. Finally, [4] use policy gradient algorithms for non-rectangular robust MDPs. While this work presents nice theoretical guarantees of convergence, there is no practical Deep RL algorithms for learning optimal robust policies.

**Deep Robust RL Methods.**  Many Deep Robust algorithms exist such as M2TD3 [18], M3DDPG [19], or RARL [20], which are all based on the two player zero-sum game presented in 2. We will compare our method against these algorithms, except [19] which is outperformed by [18] in general. We also compare our algorithm to Domain randomization (DR) [21] that learns a value function $V(s) = \max_\pi \mathbb{E}_{p \sim \mathcal{U}(\mathcal{P})} V_p^\pi(s)$ which maximizes the expected return on average across a fixed (generally uniform) distribution on $\mathcal{P}$. As such, DR approaches do not optimize the worst-case performance but still have good performance on average. Nonetheless, DR has been used convincingly in applications [22, 23]. Finally, the zero-sum game formulation has lead to the introduction of action robustness [9] which is a specific case of rectangular MDPs, in scenarios where the adversary shares the same action space as the agent and interferes with the agent's actions. Several strategies based on this idea have been proposed. One approach, the Game-theoretic Response Approach for Adversarial Defense (GRAD) [24] builds on the Probabilistic Action Robust MDP (PR-MDP) [9]. This method introduces time-constrained perturbations in both the action and state spaces and employs a game-theoretic approach with a population of adversaries. In contrast to GRAD, where temporal disturbances affect the transition kernel around a nominal kernel, our method is part of a broader setting in which the transition kernel is included in a larger uncertainty set. Robustness via Adversary Populations (RAP) [25] introduces a population of adversaries. This approach ensures that the agent develops robustness against a wide range of potential perturbations, rather than just a single one, which helps prevent convergence to suboptimal stationary points. Similarly, State Adversarial MDPs [26, 27, 28, 24] address adversarial attacks on state observations, effectively creating a partially observable MDP. Finally, using rectangularity assumptions, [29, 30] use Wasserstein and $\chi^2$ balls respectively for the uncertainty set in Robust RL.

## 4    Time-constrained robust MDP algorithms

The TC-RMDP framework addresses the limitations of traditional robust reinforcement learning by considering multifactorial, correlated, and time-dependent disturbances. Traditional robust reinforcement learning often relies on rectangularity assumptions, which are rarely met in real-

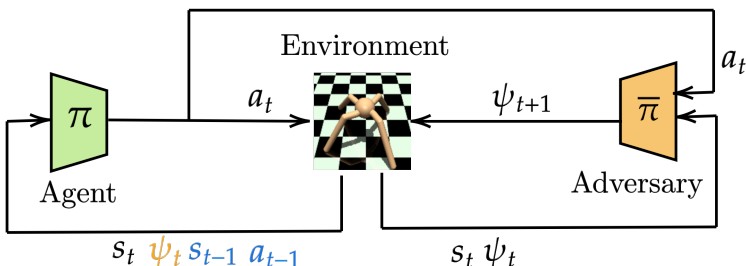

Figure 1: TC-RMDP training involves a temporally-constrained adversary aiming to maximize the effect of temporally-coupled perturbations. Conversely, the agent aims to optimize its performance against this time-constrained adversary. In orange, the oracle observation, and in blue the stacked observation.

world scenarios, leading to overly conservative policies. The TC-RMDP framework provides a more accurate reflection of real-world dynamics, moving beyond the conventional rectangularity paradigm.

We cast the TC-RMDP problem as a two-player zero-sum game, where the agent interacts with the environment, and the adversary (nature) changes the MDP parameters $\psi$. Our approach is generic and can be derived within any robust value iteration scheme, performing $\max_{\pi(s) \in \Delta_A} \min_{\psi \in \Psi} \mathbb{E}_{a \sim \pi(s)}[r(s,a) + \gamma \mathbb{E}_{s' \sim p_\psi(s,a)} v_n(s')]$ updates, by modifying the adversary's action space and potentially the agent's state space to obtain updates of the form $\max_{\pi(s,\psi) \in \Delta_A} \min_{b \in B} \mathbb{E}_{a \sim \pi(s)}[r(s,a) + \gamma \mathbb{E}_{s' \sim p_{\psi+b}(s,a)} v_n(s')]$. In Section 5, we will introduce time constraints within two specific robust value iteration algorithms, namely RARL [20] and M2TD3 [18] by simply limiting the search space for worst-case $\psi$ at each step. This specific implementation extends the original actor-critic algorithms. For the sake of conciseness, we refer the reader to Appendix E.1 for details regarding the loss functions and algorithmic details.

Three variations of the algorithm are provided (illustrated in Figure 1) but all fall within the training loop of Algorithm 1.

---

**Algorithm 1** Time-constrained robust training

---

**Input:** Time-constrained MDP: $(S, A, \Psi, p_\psi, r, L)$, Agent $\pi$, Adversary $\bar{\pi}$
**for** *each interaction time step t* **do**
$\quad a_t \sim \pi_t(s_t, \psi_t)$          // Sample an action with Oracle-TC
$\quad$ or $a_t \sim \pi_t(s_t, a_{t-1}, s_{t-1})$      // Sample an action with Stacked-TC
$\quad$ or $a_t \sim \pi_t(s_t)$             // Sample an action with TC
$\quad \psi_{t+1} \sim \bar{\pi}_t(s_t, a_t, \psi_t)$      // Sample the worst TC parameter
$\quad s_{t+1} \sim p_{\psi_{t+1}}(s_t, a_t)$         // Sample a transition
$\quad \mathcal{B} \leftarrow \mathcal{B} \cup \{(s_t, a_t, r(s_t, a_t), \psi_t, \psi_{t+1}, s_{t+1})\}$   // Add transition to replay buffer
$\quad \{s_i, a_i, r(s_i, a_i), \psi_i, \psi_{i+1}, s_{i+1}\}_{i \in [1,N]} \sim \mathcal{B}$   // Sample a mini-batch of transitions
$\quad \pi_{t+1} \leftarrow UpdatePolicy(\pi_t)$      // Update Agent
$\quad \bar{\pi}_{t+1} \leftarrow UpdatePolicy(\bar{\pi}_t)$      // Update Adversary

---

Oracle-TC . As discussed in Section 2, the Oracle-TC version includes the MDP state and parameter value as input, $\pi : \mathcal{S} \times \Psi \to \mathcal{A}$. This method assumes that the agent has access to the true parameters of the environment, allowing it to make the most informed decisions and possibly reach the true robust value function. However, these parameters $\psi$ are sometimes non-observable in practical scenarios, making this method not always feasible.

Stacked-TC . Since $\psi$ might not be observable but may be approximately identified by the last transitions, the Stacked-TC policy uses the previous state and action as additional inputs in an attempt to replace $\psi$, $\pi : \mathcal{S} \times \mathcal{A} \times \mathcal{S} \to \mathcal{A}$. This approach leverages the information in the transitions, even though it might be insufficient for a perfect estimate of $\psi$. It aims to retain (approximately) the convergence properties of the Oracle-TC algorithm.

`Vanilla TC` . Finally, the vanilla `TC` version takes only the state, $\pi : \mathcal{S} \to \mathcal{A}$, as input, similar to standard robust MDP policies. This method does not attempt to infer the environmental parameters or the transition dynamics explicitly. Instead, it relies on the current state information to guide the agent's actions. While this version is the most straightforward and computationally efficient, it may not perform as robustly as the `Oracle-TC` or `Stacked-TC` versions in environments with significant temporal disturbances, since it attempts to solve a partially observable Markov game, for which there may not exist a stationary optimal policy based only on the observation. Despite this, it remains a viable option in scenarios where computational simplicity and quick decision-making are prioritized.

## 5 Results

**Experimental settings**. This section evaluates the robust time-constrained algorithm's performance under severe time constraints and in the static settings. Experimental validation was conducted in continuous control scenarios using the MuJoCo simulation environments [5]. The approach was categorized into three variants. The `Oracle-TC` , where the agent accessed environmental parameters $\pi(s_t, \psi)$; the `Stacked-TC` , where the agent took in input $\pi(s_t, s_{t-1}, a_{t-1})$; and the vanilla `TC` , which did not receive any additional inputs $\pi(s)$. For each variant of the time-constrained algorithms, we applied them to RARL [20], and M2TD3 [18], renaming them TC-RARL and TC-M2TD3, respectively. The algorithms were tested against two state-of-the-art robust reinforcement learning algorithms, M2TD3 and RARL. Additionally, the Oracle versions of M2TD3 and RARL, where the agent's policy included $\psi$ in the input $\pi : \mathcal{S} \times \Psi \to \mathcal{A}$, were evaluated for a more comprehensive assessment. Comparisons were also made with Domain Randomization (DR) [21] and vanilla TD3. [31] to ensure a thorough analysis. A 3D uncertainty set is defined in each environment $\mathcal{P}$ normalized between $[0, 1]^3$. Appendix G provides detailed descriptions of uncertainty parameters. Performance metrics were gathered after five million steps to ensure a fair comparison. All baselines were constructed using TD3, and a consistent architecture was maintained across all TD3 variants. The results presented below were obtained by averaging over ten distinct random seeds. Appendices E.4, E.3, E.2, and H.2 discuss further details on hyperparameters, network architectures, and implementation choices, including training curves for our methods and baseline comparisons. In the following tables 1, 2, 3, the best performances are shown in bold. Oracle methods, with access to optimal information, are shown in black. Items in bold and green represent the best performances with limited information on $\psi$, making them more easily usable in many scenarios. When there is only one element in bold and green, this implies that the best overall method is a non-oracle method.

| | Ant | HalfCheetah | Hopper | Humanoid | Walker | Agg. |
|---|---|---|---|---|---|---|
| Oracle M2TD3 | $1.11 \pm 0.07$ | $0.95 \pm 0.1$ | $1.51 \pm 0.84$ | $2.07 \pm 0.19$ | $1.31 \pm 0.36$ | $1.39 \pm 0.31$ |
| Oracle RARL | $0.72 \pm 0.18$ | $-0.71 \pm 0.05$ | $-1.3 \pm 0.28$ | $-2.8 \pm 1.62$ | $-0.19 \pm 0.2$ | $-0.86 \pm 0.47$ |
| `Oracle-TC` -M2TD3 | $1.61 \pm 0.32$ | $\mathbf{2.76 \pm 0.16}$ | $\mathbf{7.79 \pm 1.0}$ | $1.69 \pm 2.14$ | $1.49 \pm 0.41$ | $\mathbf{3.07 \pm 0.81}$ |
| `Oracle-TC` -RARL | $\mathbf{1.66 \pm 0.32}$ | $2.63 \pm 0.12$ | $6.86 \pm 1.46$ | $0.19 \pm 1.68$ | $1.34 \pm 0.11$ | $2.54 \pm 0.74$ |
| `Stacked-TC` -M2TD3 | $1.33 \pm 0.21$ | $\mathbf{\color{green}{2.4 \pm 0.19}}$ | $\mathbf{\color{green}{6.51 \pm 0.59}}$ | $-1.42 \pm 1.44$ | $\mathbf{\color{green}{1.69 \pm 0.33}}$ | $2.1 \pm 0.55$ |
| `Stacked-TC` -RARL | $1.48 \pm 0.22$ | $1.76 \pm 0.08$ | $3.28 \pm 0.27$ | $1.39 \pm 0.57$ | $1.01 \pm 0.21$ | $1.78 \pm 0.27$ |
| `TC` -M2TD3 | $1.52 \pm 0.2$ | $\mathbf{\color{green}{2.42 \pm 0.1}}$ | $5.16 \pm 0.2$ | $\mathbf{\color{green}{4.02 \pm 1.23}}$ | $1.38 \pm 0.25$ | $\mathbf{\color{green}{2.9 \pm 0.4}}$ |
| `TC` -RARL | $1.57 \pm 0.26$ | $1.54 \pm 0.15$ | $2.04 \pm 0.49$ | $1.25 \pm 1.91$ | $0.89 \pm 0.2$ | $1.46 \pm 0.6$ |
| TD3 | $0.0 \pm 0.19$ | $0.0 \pm 0.27$ | $0.0 \pm 1.27$ | $0.0 \pm 1.18$ | $0.0 \pm 0.23$ | $0.0 \pm 0.63$ |
| DR | $\mathbf{\color{green}{1.58 \pm 0.2}}$ | $1.59 \pm 0.12$ | $2.28 \pm 0.42$ | $0.87 \pm 1.79$ | $1.03 \pm 0.19$ | $1.47 \pm 0.54$ |
| M2TD3 | $1.0 \pm 0.19$ | $1.0 \pm 0.14$ | $1.0 \pm 0.96$ | $1.0 \pm 1.31$ | $1.0 \pm 0.31$ | $1.0 \pm 0.58$ |
| RARL | $0.63 \pm 0.2$ | $-0.61 \pm 0.18$ | $-1.5 \pm 0.33$ | $0.8 \pm 0.88$ | $0.27 \pm 0.25$ | $-0.08 \pm 0.37$ |

Table 1: Avg. of normalized time-coupled worst-case performance over 10 seeds for each method

**Performance of TCRMDPs in worst-case time-constrained**. Table 1 reports the worst-case time-constrained perturbation. To address the worst-case time-constrained perturbations for each trained agent $\pi^*$, we utilized a time-constrained adversary using TD3 algorithm $\bar{\pi}^* = \min_{b \in B} \mathbb{E}_{a \sim \pi^*(s), b \sim \bar{\pi}(s, a, \psi)}[r(s, a) + \gamma \mathbb{E} s' \sim p_{\psi+b}(s, a) v_n(s')]$ within a perturbation radius of $L = 0.001$ for a total of 5 million steps. The sum of episode rewards was averaged over 10 episodes. A heuristic to choose the perturbation radius is explained in Appendix K. To compare metrics across different environments, each method's score $v$ was standardized relative to the reference score of TD3. TD3 was trained on the environment using default transition function parameters, with its score denoted as $v_{TD3}$. The M2TD3 score, $v_{M2TD3}$, was used as the comparison target. The formula applied was $(v - v_{TD3})/(|v_{M2TD3} - v_{TD3}|)$. This positioned $v_{TD3}$ as the minimal baseline and $v_{M2TD3}$ as the target score. This standardisation provides a metric that quantifies the

improvement of each method over TD3 in relation to the improvement of M2TD3 over TD3. In each evaluation environment, agents trained with the time-constrained framework (indicated by TC in the method name) demonstrated significantly superior performance compared to those trained using alternative robust reinforcement learning approaches, including M2TD3 and RARL. Furthermore, they outperformed those trained through domain randomisation (DR). Notably, even without directly conditioning the policy with $\psi$, the time-constrained trained policies excelled against all baselines, achieving up to a 2.9-fold improvement. The non-normalized scores are reported in Appendix H. Additionally, when policies were directly conditioned by $\psi$ and trained within the robust reinforcement learning framework, they tended to be overly conservative in the time-constrained framework. This is depicted in Table 1, comparing the performances of Oracle RARL, Oracle M2TD3, Oracle TC-RARL, and Oracle TC-M2TD3. Both policies also observe $\psi$. The only difference is that Oracle RARL and Oracle M2TD3 were trained in the robust reinforcement learning framework, while Oracle TC-RARL and Oracle TC-M2TD3 were trained in the time-constrained framework. The performance differences under worst-case time-coupled perturbation are as follows: for Oracle RARL (resp. M2TD3) and Oracle TC-RARL (resp. M2TD3), the values are $-0.86$ (1.39) vs. 2.54 (3.07). This observation highlights the need for a balance between robust training and flexibility in dynamic conditions. A natural question arises regarding the worst-case time-constrained perturbation. Was the adversary in the loop adequately trained, or might its suboptimal performance lead to overestimating the trained agent's reward against the worst-case perturbation? The adversary's performance was monitored during its training against all fixed-trained agents. The results in Appendix F show that our adversary converged.

**Robust Time-Constrained Training under various time fixed adversaries**. The method was evaluated against various fixed adversaries, focusing on the random fixed adversary shown in Figure 2. This evaluation shows that robustly trained agents can handle dynamic and unpredictable conditions. The random fixed adversary simulates stochastic changes by selecting a parameter $\psi_t$ at each timestep within a radius of $L = 0.1$. This radius is 100 times larger than in our training methods. At the start of each episode, $\psi_0$ is uniformly sampled from the uncertainty set $\psi_0 \sim \mathcal{U}(\mathcal{P})$. This tests the agents' adaptability to unexpected changes. Figures 2a through 2e show our agents' performance. Agents trained with our robust framework consistently outperformed those trained with standard methods. The policy was also assessed against five other fixed adversaries: cosine, exponential, linear, and logarithmic. Detailed results are provided in the Appendix. H.1.

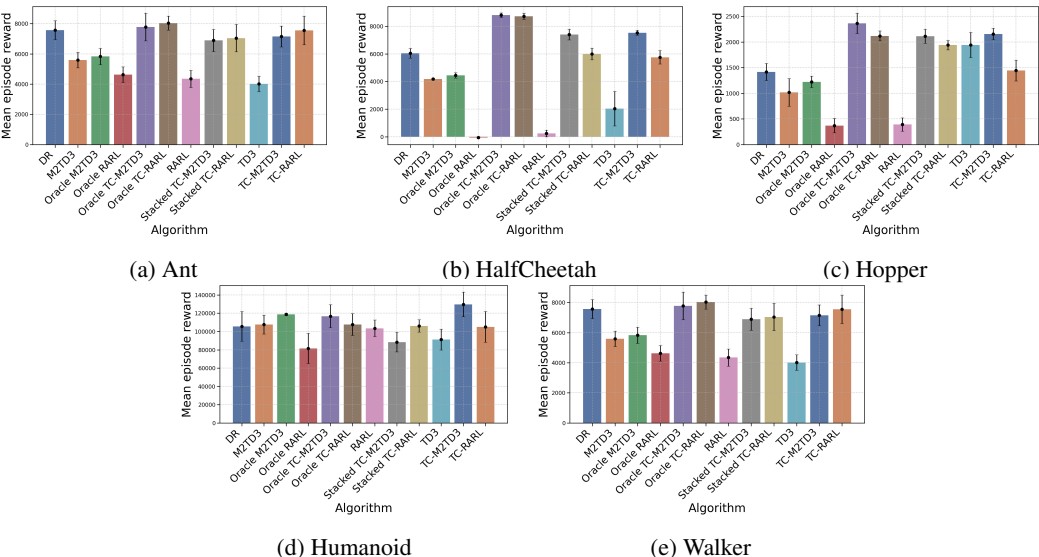

Figure 2: Evaluation against a random fixed adversary, with a radius $L = 0.1$

**Performance of Robust Time-Constrained MDPs in the static setting**. In static environments, the Robust Time-Constrained algorithms were evaluated for worst-case and average performance metrics, shown in Tables 2 and 3. A fixed uncertainty set $\mathcal{P}$ was used, dividing each dimension of $\Psi$ into ten segments, creating a grid of 1000 points ($10^3$). Each agent ran five episodes at each grid point, and the rewards were averaged. The scores were normalized as described for the

time-constrained adversary analysis in Table 1. The raw data is provided in Appendix 9 and 10. Performance scores were adjusted relative to the baseline $v_{TD3}$ and $v_{M2TD3}$. As a result, normalized results reveal distinct trends among agent configurations within the TC-RMDP framework. The Oracle TC-M2TD3 variant achieved an average score of 3.12 3, while the Stacked TC-M2TD3 scored 2.23, indicating its resilience. Furthermore, in the worst-case scenario, the TC-RARL and Stacked TC-RARL variants demonstrated adaptability, with TC-RARL scoring 0.92 and TC-M2TD3 scoring 1.02 2. This performance highlights its reliability in challenging static environments.

| | Ant | HalfCheetah | Hopper | Humanoid | Walker | Agg |
|---|---|---|---|---|---|---|
| Oracle M2TD3 | **1.02 ± 0.19** | 0.34 ± 0.23 | 0.97 ± 0.55 | **3.9 ± 3.65** | 0.3 ± 0.45 | **1.31 ± 1.01** |
| Oracle RARL | 0.62 ± 0.32 | 0.1 ± 0.02 | 0.48 ± 0.19 | −2.59 ± 2.18 | 0.16 ± 0.21 | −0.25 ± 0.58 |
| Oracle-TC -M2TD3 | 0.1 ± 0.25 | **1.87 ± 0.1** | 0.49 ± 1.07 | −0.8 ± 3.05 | 0.28 ± 0.38 | 0.39 ± 0.97 |
| Oracle-TC -RARL | 0.59 ± 0.36 | 1.55 ± 0.35 | 0.4 ± 0.16 | 1.19 ± 1.24 | 0.56 ± 0.39 | 0.86 ± 0.5 |
| Stacked-TC -M2TD3 | −0.05 ± 0.09 | **1.56 ± 0.16** | 1.08 ± 0.89 | −0.83 ± 2.62 | 1.12 ± 0.5 | 0.58 ± 0.85 |
| Stacked-TC -RARL | 0.07 ± 0.13 | 0.76 ± 0.34 | **1.35 ± 0.93** | **1.75 ± 2.48** | 0.67 ± 0.32 | 0.92 ± 0.84 |
| TC -M2TD3 | −0.06 ± 0.08 | 1.49 ± 0.23 | 1.29 ± 0.29 | 1.21 ± 2.44 | **1.19 ± 0.34** | **1.02 ± 0.68** |
| TC -RARL | 0.14 ± 0.24 | 0.89 ± 0.3 | 1.5 ± 0.76 | 1.4 ± 4.57 | 0.67 ± 0.59 | 0.92 ± 1.29 |
| TD3 | 0.0 ± 0.34 | 0.0 ± 0.06 | 0.0 ± 0.21 | 0.0 ± 2.27 | 0.0 ± 0.1 | 0.0 ± 0.6 |
| DR | 0.06 ± 0.16 | 1.07 ± 0.36 | 0.86 ± 0.82 | 0.04 ± 4.1 | 0.57 ± 0.37 | 0.52 ± 1.16 |
| M2TD3 | **1.0 ± 0.27** | 1.0 ± 0.16 | 1.0 ± 0.65 | 1.0 ± 3.32 | 1.0 ± 0.63 | 1.0 ± 1.01 |
| RARL | 0.44 ± 0.3 | 0.13 ± 0.08 | 0.5 ± 0.22 | 0.44 ± 2.94 | 0.12 ± 0.09 | 0.33 ± 0.73 |

Table 2: Avg. of normalized static worst-case performance over 10 seeds for each method

| | Ant | HalfCheetah | Hopper | Humanoid | Walker | Agg |
|---|---|---|---|---|---|---|
| Oracle M2TD3 | 1.13 ± 0.08 | 1.56 ± 0.24 | 1.12 ± 0.46 | 1.96 ± 1.53 | 1.23 ± 0.3 | 1.4 ± 0.52 |
| Oracle RARL | 0.7 ± 0.22 | −1.4 ± 0.13 | −0.77 ± 0.24 | −2.6 ± 2.88 | −1.13 ± 0.84 | −1.04 ± 0.86 |
| Oracle-TC -M2TD3 | 1.73 ± 0.09 | **4.35 ± 0.26** | **5.54 ± 0.13** | 2.12 ± 1.4 | 1.84 ± 0.37 | **3.12 ± 0.45** |
| Oracle-TC -RARL | **1.78 ± 0.02** | 4.32 ± 0.21 | 5.08 ± 0.48 | 0.42 ± 2.9 | 1.68 ± 0.24 | 2.66 ± 0.77 |
| Stacked-TC -M2TD3 | 1.45 ± 0.38 | **3.78 ± 0.29** | **5.2 ± 0.29** | −1.38 ± 1.67 | **2.11 ± 0.52** | 2.23 ± 0.63 |
| Stacked-TC -RARL | 1.52 ± 0.11 | 2.29 ± 0.23 | 2.91 ± 0.67 | 1.14 ± 2.19 | 1.21 ± 0.46 | 1.81 ± 0.73 |
| TC -M2TD3 | 1.6 ± 0.06 | 3.71 ± 0.24 | 4.4 ± 0.6 | **3.28 ± 2.52** | 1.56 ± 0.23 | **2.91 ± 0.73** |
| TC -RARL | **1.67 ± 0.07** | 2.27 ± 0.22 | 1.79 ± 0.53 | 0.89 ± 2.19 | 1.01 ± 0.21 | 1.53 ± 0.64 |
| TD3 | 0.0 ± 0.49 | 0.0 ± 0.22 | 0.0 ± 0.83 | 0.0 ± 1.36 | 0.0 ± 0.51 | 0.0 ± 0.68 |
| DR | 1.65 ± 0.05 | 2.31 ± 0.27 | 2.08 ± 0.49 | 1.15 ± 2.47 | 1.22 ± 0.34 | 1.68 ± 0.72 |
| M2TD3 | 1.0 ± 0.11 | 1.0 ± 0.19 | 1.0 ± 0.55 | 1.0 ± 1.43 | 1.0 ± 0.65 | 1.0 ± 0.59 |
| RARL | 0.69 ± 0.13 | −1.3 ± 0.54 | −0.99 ± 0.11 | 0.47 ± 1.92 | −0.35 ± 0.83 | −0.3 ± 0.71 |

Table 3: Avg. of normalized static average case performance over 10 seeds for each method

# 6 Some Theoretical properties of TC-MDPS.

## 6.1 On the optimal policy of TC

Following Lemma 3.3 of [1], it is known that in the rectangular case, there exists an optimal policy of the adversary that is stationary, provided that the actor policy is stationary. The TC-RMDP definition enforces a limitation on the temporal variation of the transition kernel. Consequently, all stationary adversarial policies are constrained by this stipulation. In turn, this guarantees that (under the hypothesis of $sa$-rectangularity) there always exists a solution to the TC-RMDP that is also a solution to the original RMDP. In other words: optimizing policies for TC-RMDPs do not exclude optimal solutions to the underlying RMDP. This sheds an interesting light on the search for robust optimal policies, since TC-RMDPs shrink the search space of optimal adversarial policies. In practice, this is confirmed by the previous experimental results (Figure 2) where the optimal agent policy found by either Oracle-TC , Stacked-TC , or vanilla TC actually outperforms the one found by M2TD3 or RARL in the non time-constrained setting.

## 6.2 Some Lipchitz-properties for non-stationary TC-RMPDS.

In this subsection we slightly depart from the framework defined in Section 2 and study the smoothness of the robust objective for vanilla TC or Stacked-TC . Th. 2.1 is no longer applicable as $\psi$ is not observed. However, we can still give smoothness of the objective starting from Lipchichz conditions

on the evolution of the parameter that leads to smoothness on reward and transition kernel in the following definition 6.1.

**Definition 6.1** (Reward/Kernel Lipchitz TC-RMDPs [13]). *We say that a parametric RDMPs is time constrained if the parameter change is bounded through time ie.* $\|\psi_{t+1} - \psi_t\| \leq L$. *Moreover, we assume that this variation in parameter implies a variation in the reward and transition kernel of*

$$\forall s \in \mathcal{S}, \forall a \in \mathcal{A}, \|P_t(\cdot \mid s,a) - P_{t+1}(\cdot \mid s,a)\|_1 \leq L_P \quad ; |r_t(s,a) - \mathbb{E}\left[r_{t+1}(s,a)\right]| \leq L_r$$

From a theoretical point of view, a TC-RMDP can be seen as a sequence of stationary MDPs with time indexed reward and transition kernel $r_t$, $P_t$ that have continuity. More formally for $M_t = (S, A, \Psi, p_{\psi_t}, r_t, L = (L_p, L_r))$, we can then define the sequence of stationary MDPs with Lipchitz variation :

$$\mathcal{M}_t^L = \Big\{ \{M_t\}_{t'=t_0}^t ; \exists L_r \in \mathbb{R} \forall s \in \mathcal{S}, \forall a \in \mathcal{A}, \|P_t'(\cdot \mid s,a) - P_{t'+1}(\cdot \mid s,a)\|_1 \leq L_P \quad ;$$
$$|r_{t'}(s,a) - r_{t'+1}(s,a)| \leq L_r \Big\} \tag{1}$$

Defining $r_t^k$ as the random variable corresponding to the reward function at time step $t$ for stationary MDPs, but iterating with index $k$, the stationary rollout return at time $t$ is $G(\pi, M_t) = \sum_{k \geq 0} \gamma^k r_t^k$. Assuming that at a fixed $t$ the reward and transition kernel $r_t, P_t$ are fixed, the robust objective function is:

$$J^R(\pi, t) := \min_{m=\{m_t'\}_{t'=t_0}^t \in \mathcal{M}_t^L} \mathbb{E}\left[G\left(\pi, m\right)\right]$$

This leads to the following guarantee for vanilla `TC` and `Stacked-TC` algorithms.

**Theorem 6.2.** *Assume TC-RMPDS with* $L = (L_r, L_P)$ *smoothness. Then* $\forall t \in \mathbb{N}, r_t \in [0,1]$,

$$\forall t \in \mathbb{N}^+, \forall t_0 \in \mathbb{N}^+, \quad |J^R(\pi, t_0) - J^R(\pi, t_0 + t)| \leq L't.$$

*with* $L' := \left(\frac{\gamma}{(1-\gamma)^2}L_P + \frac{1}{1-\gamma}L_r\right)$

This theorem states that a small variation of the Kernel and reward function will not affect too much the robust objective. In other terms, despite the fact that the TC Bellman operator may not admit a fixed point and yield a non-stationary sequence of value functions, variations of the expected return remain bounded. Proof of the Th. 6.2 can be found in Appendix C.

## 7 Conclusion

This paper presents a novel framework for robust reinforcement learning, which addresses the limitations of traditional methods that rely on rectangularity assumptions. These assumptions often result in overly conservative policies, which are not suitable for real-world applications where environmental disturbances are multifactorial, correlated, and time-constrained. In order to overcome these challenges, we proposed a new formulation, the Time-Constrained Robust Markov Decision Process (TC-RMDP). The TC-RMDP framework is capable of accurately capturing the dynamics of real-world environments, due to its consideration of the temporal continuity and correlation of disturbances. This approach resulted in the development of three algorithms: The three algorithms, `Oracle-TC`, `Stacked-TC`, vanilla `TC` which differ in the extent to which environmental information is incorporated into the decision-making process. A comprehensive evaluation of continuous control benchmarks using MuJoCo environments has demonstrated that the proposed TC-RMDP algorithms outperform traditional robust RL methods and domain randomization techniques. These algorithms achieved a superior balance between performance and robustness in both time-constrained and static settings. The results confirmed the effectiveness of the TC-RMDP framework in reducing the conservatism of policies while maintaining robustness. Moreover, we provided theoretical guaranties for `Oracle-TC` in Th. 2.1 and for `Stacked-TC` and vanilla `TC` in Th. 6.2. This study contributes to the field of robust reinforcement learning by introducing a time-constrained framework that more accurately reflects the dynamics observed in real-world settings. The proposed algorithms and theoretical contributions offer new avenues for the development of more effective and practical RL applications in environments with complex, time-constrained uncertainties.

## Acknowledgments

The authors acknowledge the support of the DEEL and ENVIA projects, the funding of the AI Interdisciplinary Institute ANITI funding, through the French "Investing for the Future – PIA3" program under grant agreement ANR-19-PI3A-0004. This work benefited from computing resources from CALMIP under grant P21001. Pierre Clavier has been supported by a grant from Region Ile-de-France; DIM Math Innov and also by Fondation Mathématique Jacques Hadamard.

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

# A  Appendix

The Appendix is structured as follow :

- In Appendix B, proof for fix point of `Oracle-TC` algorithm for can be found.
- In Appendix C, proof for algorithm Vanilla `TC` and `Stacked-TC` can found about robust objective.
- In Appendix F, the adversary training was sanity-checked within the time-constrained evaluation.
- In Appendix E, all implementation details are provided.
- In Appendix H, all raw results are presented.
- In Appendix I, the computer resources and training wall clock time are detailed.
- In Appendix J, the broader impact and limitations are discussed.
- In Appendix K, a heuristic for choosing the radius $L$

# B  Proof of Theorem 2.1

*Proof.* The Proof is similar to [1], using the fact that $p_{\psi+b}$ belongs to the simplex, we get contraction of the operator and convergence to a fix point $v_B^*$. Not that to converge to the fix point, there is no need of rectangularity. □

Recall the recursion

$$v_{n+1}(s,\psi) = \max_{\pi(s,\psi)\in\Delta_A} \min_{b\in B} T_b^\pi v_n(s,\psi) := \max_{\pi(s,\psi)\in\Delta_A} \min_{b\in B} \mathbb{E}_{a\sim\pi(s)}[r(s,a) + \gamma\mathbb{E}_{s'\sim p_{\psi+b}}v_n(s',\psi')] \tag{2}$$

First we prove that the TC Robust Operator $T_B^\pi$ is a contraction. Let $V_1, V_2 \in \mathbb{R}^n$. Fix $s \in S$, and assume that $T_B^\pi V_1(s,\psi) \geq T_B^\pi V_2(s,\psi)$. Then fix $\epsilon > 0$ and pick $\pi$ s.t given $s \in S$,

$$\inf_{b\in B} \mathbb{E}_{p_{\psi+b}} [r(s,\pi(s)) + \gamma V_1(s',\psi')] \geq T_B^\pi V_1(s,\psi') - \epsilon. \tag{3}$$

First we pick a probability measure $p'$ such that $p' = p_{\psi+b}, b \in B$, such that

$$\mathbb{E}_{p'}[r(s,\pi(s)) + \gamma V_2(s',\psi')] \leq \inf_{b\in B} \mathbb{E}_{p'}[r(s,\pi(s)) + \gamma V_2(s',\psi')] + \epsilon. \tag{4}$$

Then it lead to

$$0 \leq T_B^\pi V_1(s,\psi) - T_B^\pi V_2(s,\psi) \leq \left(\inf_{p\in B} \mathbb{E}_p[r(s,\pi(s)) + \gamma V_1(s',\psi')] + \epsilon\right) \tag{5}$$

$$- \left(\inf_{p\in B} \mathbb{E}_p[r(s,\pi(s)) + \gamma V_2(s',\psi')]\right) \tag{6}$$

$$\leq (\mathbb{E}_{p'}[r(s,\pi(s)) + \gamma V_1(s',\psi')] + \epsilon) - \tag{7}$$

$$(\mathbb{E}_{p'}[r(s,\pi(s)) + \gamma V_2(s',\psi')] - \epsilon), \tag{8}$$

$$= \gamma\mathbb{E}_{p'}[V_1 - V_2] + 2\epsilon, \tag{9}$$

$$\leq \gamma\mathbb{E}_{p'}|V_1 - V_2| + 2\epsilon \tag{10}$$

$$\leq \gamma\|V_1 - V_2\|_\infty + 2\epsilon. \tag{11}$$

where last inequality is Holder's inequality between $L_1$ and $L_\infty$ norms, use probability measure in the simplex such as $\|p'\|_1 = 1$. Doing the same thing but in the case where $T_B^\pi V_1(s) \leq T_B^\pi V_2(s)$, it holds

$$\forall s \in S, |T_B^\pi V_1(s) - T_B^\pi V_2(s)| \le \gamma \|V_1 - V_2\|_\infty + 2\epsilon, \tag{12}$$

i.e. $\|T_B^\pi V_1 - T_B^\pi V_2\|_\infty \le \gamma \|V_1 - V_2\|_\infty + 2\epsilon$. As we can choose $\epsilon$ arbitrary small, this establishes that the TC Bellman operator is a $\gamma$-contraction. Since $T_B^\pi$ is a contraction operator on a Banach space, the Banach fixed point theorem implies that the operator equation $T_B^\pi V = V$ has a unique solution $V = v_B^\pi$. A similar proof can be done for optimal operator $T_B^*$. The only difference is the maximum operator which is $1-$Lipschitz. So $T_B^*$ is also a contraction. Then, once proved that operators are $\gamma-$ contraction, following [1] (Th. 5), we have that the fixed point of this recursion is exactly :

$$v_B^\pi(s, \psi) := \min_{\substack{(b_t)_{t \in \mathbb{N}}, \\ b_t \in B}} \mathbb{E}\Big[ \sum \gamma^t r_t | \psi_{-1} = \psi, s_0 = s, b_t \in B, \psi_t = \psi_{t-1} + b_t, a \sim \pi, s_t \sim p_{\psi_t} \Big], \tag{13}$$

$$v_B^*(s, \psi) = \max_{\pi(s,\psi) \in \Delta_A} v_B^\pi(s, \psi). \tag{14}$$

for (optimal) TC Bellman Operator.

## C  Guaranties for non-stationary Robust MDPS

Recall that we represent a non-stationary robust MDPs (NS-RMDP) as a stochastic sequence, $\{\mathcal{M} = \{M_i\}_{t=t_0}^\infty$, of stationary MDPs $M_t \in \mathcal{M}$, where $\mathcal{M}$ is the set of all stationary MDPs. Each $M_t$ is a tuple, $(\mathcal{S}, \mathcal{A}, P_t, r_t, \gamma, \rho^0)$, where $\mathcal{S}$ The set of possible states is denoted by $\mathcal{S}$, the set of actions by $\mathcal{A}$, the discounting factor by $\gamma$, the start-state distribution by $\rho^0$, and the reward distribution by $r_t$. The reward distribution, denoted by $r_t : \mathcal{S} \times \mathcal{A} \to \Delta(\mathbb{R})$, is the probability distribution of rewards. The transition function, represented by $P_t : \mathcal{S} \times \mathcal{A} \to \Delta(\mathcal{S})$, is the probability distribution of transitions between states. The symbol $\Delta$ denotes the simplex. For all $M_t \in \mathcal{M}$, we assume that the state space, action space, discount factor, and initial distribution remain fixed. A policy is represented as a function $\pi : \mathcal{S} \to \Delta(\mathcal{A})$. In general, we will use subscripts $t$ to denote the time evolution during an episode and superscripts $k$ to denote the time step assuming reward or kernel $t$ which is stationary, assuming that the reward function is not changing as it is at time step $t$ stationary. That $r_t^k$ is the random variables corresponding to the state, action, and reward at time step $t$ for stationary, but iterating with index $k$.

**Definition C.1** ( Lipschitz of sequence of MDPs). *We denote the sequence of kernel and reward function $\mathcal{P} = \{P_t\}_{t=t_0}^\infty$ and $\mathcal{R} = \{r_t\}_{t=t_0}^\infty$. We define a sequence of MDP is $L = (L_r, L_P)$-Lipschitz if $m = \{m_t\}_{t=t_0}^\infty \in \mathcal{M}^L$ with*

$$\mathcal{M}_t^L = \Big\{ \{M_t\}_{t'=t_0}^t ; \exists (L_r, L_P) \in \mathbb{R}_+^2 \forall t \in \mathbb{N}, \forall s \in \mathcal{S}, \forall a \in \mathcal{A}, \|P_{t'}(\cdot \mid s, a) - P_{t'+1}(\cdot \mid s, a)\|_1 \le L_P$$

$$; |r_t'(s, a) - r_{t'+1}(s, a)| \le L_r \Big\}$$

Assuming that for a time steps the reward function is stationary, we can compute the average return as:

**Definition C.2.** *Non-robust objective function, assuming that $G(\pi, M_t) = \sum_{k \ge 0} \gamma^k r_t^k$, the return is we assume stationary with reward function $r_t$*

$$J(\pi, t) = \mathbb{E}[G(\pi, M_t)] = (1 - \gamma)^{-1} \sum_{s \in \mathcal{S}} d^\pi(s, M_t) \sum_{a \in \mathcal{A}} \pi(a \mid s) r_t(s, a). \tag{15}$$

*with $d^\pi$ the state occupancy measure defined in* (16).

**Definition C.3** (Robust (optimal) Return of NS-RMDPs). *Let a return of $\pi$ for any $m_t \in M_t$ be $G(\pi, M_t) := \sum_{k=0}^\infty \gamma^k r_t^k$ with kernel transition $P_t$ following $\pi$, with $\forall k, t, r_t^k \in [0, 1]$, and the Robust non-stationary expected return with variation of kernel*

*Let the robust performance of $\pi$ for episode $t$ be*

$$J^R(\pi, t) := \min_{m = \{m_t'\}_{t'=t_0}^t \in \mathcal{M}_t^L} \mathbb{E}[G(\pi, m)]$$

# D   Proof Theom 6.2

$$\forall t \in \mathbb{N}^+, \forall t_0 \in \mathbb{N}^+, \quad |J^R(\pi, t_0) - J^R(\pi, t_0 + t)| \leq L't.$$

with $L' := \left( \frac{\gamma}{(1-\gamma)^2} L_P + \frac{1}{1-\gamma} L_r \right)$

*Proof of Theorem 6.2.* First, this difference can be upper bounded in the non robust case as:
By definition, we can rewrite non-robust objective function and occupancy measure as.

$$d^\pi(s, M_t) = (1-\gamma) \sum_{k=0}^{\infty} \gamma^k \Pr(S_t = s \mid \pi, M_t), \tag{16}$$

$$J(\pi, M_t) = (1-\gamma)^{-1} \sum_{s \in \mathcal{S}} d^\pi(s, M_t) \sum_{a \in \mathcal{A}} \pi(a \mid s) r_t(s, a). \tag{17}$$

First, we can decompose the problem into sub-problems such that

$$\forall t \in \mathbb{N}^+, \forall t_0 \in \mathbb{N}^+, \quad |J(\pi, t_0) - J(\pi, t_0 + t)| \leq \left| \sum_{t'=t_0}^{t_0+t-1} |J(\pi, M_{t'}) - J(\pi, M_{t'+1})| \right. \tag{18}$$

using triangular inequality. Looking at differences between two time steps:

$$(1-\gamma)|J(\pi, M_t) - J(\pi, M_{t+1})|$$

$$= \left| \sum_{s \in \mathcal{S}} d^\pi(s, M_t) \sum_{a \in \mathcal{A}} \pi(a \mid s) r_t(s, a) - \sum_{s \in \mathcal{S}} d^\pi(s, M_{t+1}) \sum_{a \in \mathcal{A}} \pi(a \mid s) r_{t+1}(s, a) \right|$$

$$= \left| \sum_{s \in \mathcal{S}} \sum_{a \in \mathcal{A}} \pi(a \mid s) \left( d^\pi(s, M_t) r_t(s, a) - d^\pi(s, M_{t+1}) r_{t+1}(s, a) \right) \right|$$

$$= \left| \sum_{s \in \mathcal{S}} \sum_{a \in \mathcal{A}} \pi(a \mid s) \left( d^\pi(s, M_t) (r_{t+1}(s, a) + (r_t(s, a) - R_{t+1}(s, a))) - d^\pi(s, M_{t+1}) r_{t+1}(s, a) \right) \right|$$

$$= \left| \sum_{s \in \mathcal{S}} \sum_{a \in \mathcal{A}} \pi(a \mid s) \left( d^\pi(s, M_t) - d^\pi(s, M_{t+1}) \right) r_{t+1}(s, a) \right.$$

$$\left. + \sum_{s \in \mathcal{S}} \sum_{a \in \mathcal{A}} \pi(a \mid s) d^\pi(s, M_t) (r_t(s, a) - r_{t+1}(s, a)) \right|$$

$$\overset{(a)}{\leq} \sum_{s \in \mathcal{S}} \sum_{a \in \mathcal{A}} \pi(a \mid s) |d^\pi(s, M_t) - d^\pi(s, M_{t+1})| |r_{t+1}(s, a)|$$

$$+ \sum_{s \in \mathcal{S}} \sum_{a \in \mathcal{A}} \pi(a \mid s) d^\pi(s, M_t) |r_t(s, a) - r_{t+1}(s, a)|$$

$$\overset{(b)}{\leq} \sum_{s \in \mathcal{S}} \sum_{a \in \mathcal{A}} \pi(a \mid s) |d^\pi(s, M_t) - d^\pi(s, M_{t+1})| + L_R \sum_{s \in \mathcal{S}} \sum_{a \in \mathcal{A}} \pi(a \mid s) d^\pi(s, M_t)$$

$$= \sum_{s \in \mathcal{S}} |d^\pi(s, M_t) - d^\pi(s, M_{t+1})| + L_r$$

where (a) is triangular inequality, (b) is definition of of supremum of reward in the assumptions and reward bounded by 1. Then, let $P_t^\pi \in \mathbb{R}^{|\mathcal{S}| \times |\mathcal{S}|}$ be the transition matrix ( $s'$ in rows and $s$ in columns) resulting due to $\pi$ and $P_t$, i.e., $\forall t, P_t^\pi(s', s) := \Pr(S_{t+1} = s' \mid S_t = s, \pi, M_t)$, and let $d^\pi(\cdot, M_t) \in \mathbb{R}^{|\mathcal{S}|}$ denote the vector of probabilities for each state, then Finally we can easily bound the difference of occupation measure as :

$$\sum_{s \in \mathcal{S}} |d^\pi (s, M_t) - d^\pi (s, M_{t+1})| \tag{19}$$

$$\overset{(d)}{\leq} \gamma(1-\gamma)^{-1} \sum_{s' \in \mathcal{S}} \left| \sum_{s \in \mathcal{S}} \left( P_t^\pi (s', s) - P_{t+1}^\pi (s', s) \right) d^\pi (s, M_t) \right| \tag{20}$$

$$\leq \gamma(1-\gamma)^{-1} \sum_{s' \in \mathcal{S}} \sum_{s \in \mathcal{S}} \left| P_t^\pi (s', s) - P_{t+1}^\pi (s', s) \right| d^\pi (s, M_t) \tag{21}$$

$$= \gamma(1-\gamma)^{-1} \sum_{s' \in \mathcal{S}} \sum_{s \in \mathcal{S}} \left| \sum_{a \in \mathcal{A}} \pi(a \mid s) \left( \Pr(s' \mid s, a, M_t) - \Pr(s' \mid s, a, M_{t+1}) \right) \right| d^\pi (s, M_t) \tag{22}$$

$$\leq \gamma(1-\gamma)^{-1} \sum_{s' \in \mathcal{S}} \sum_{s \in \mathcal{S}} \sum_{a \in \mathcal{A}} \pi(a \mid s) \left| \Pr(s' \mid s, a, M_t) - \Pr(s' \mid s, a, M_{t+1}) \right| d^\pi (s, M_t) \tag{23}$$

$$= \gamma(1-\gamma)^{-1} \sum_{s \in \mathcal{S}} \sum_{a \in \mathcal{A}} \pi(a \mid s) d^\pi (s, M_t) \sum_{s' \in \mathcal{S}} \left| \Pr(s' \mid s, a, M_t) - \Pr(s' \mid s, a, M_{t+1}) \right| \tag{24}$$

$$\leq \gamma(1-\gamma)^{-1} \sum_{s \in \mathcal{S}} \sum_{a \in \mathcal{A}} \pi(a \mid s) d^\pi (s, M_t) L_P \tag{25}$$

$$= \frac{\gamma L_P}{(1-\gamma)}, \tag{26}$$

which gives regrouping all terms:

$$|J(\pi, M_t) - J(\pi, M_{t+1})| \leq \frac{L_r}{1-\gamma} + \frac{\gamma L_P}{(1-\gamma)^2}. \tag{27}$$

where the stationary MDP $M_{t+1}$ can be chosen as the minimum over the previous MDPs at time step $t$ such as $|\Pr(s' \mid s, a, M_t) - \Pr(s' \mid s, a, M_{t+1})| \leq L_p$. Rewriting previous equation (27), it holds that

$$\left| \left[ \mathbb{E}_{\pi, P} [G(\pi, m)] - \min_{m = \{m'_t\}_{t'=t}^{t+1}} \mathbb{E}_{\pi, P} [G(\pi, m)] \right] \right| \leq \frac{L_r}{1-\gamma} + \frac{\gamma L_P}{(1-\gamma)^2} = L'. \tag{28}$$

Now considering non robust objective :

$$\left| J^R (\pi, t) - J^R (\pi, t+1) \right| \tag{29}$$

$$= \left| \min_{m = \{m'_t\}_{t'=t_0}^t \in \mathcal{M}^L} \mathbb{E} [G(\pi, m)] - \min_{m = \{m'_t\}_{t'=t_0}^{t+1} \in \mathcal{M}_{t+1}^L} \mathbb{E} [G(\pi, m)] \right| \tag{30}$$

$$= \left| \min_{m = \{m'_t\}_{t'=t_0}^t \in \mathcal{M}_t^L} \left[ \mathbb{E} [G(\pi, m)] - \min_{m = \{m'_t\}_{t'=t_0}^t \in \mathcal{M}_t^L} \min_{m = \{m'_t\}_{t'=t}^{t+1}} \mathbb{E} [G(\pi, m)] \right] \right| \tag{31}$$

$$\leq \max_{m = \{m_t\}_{t=t_0}^t \in \mathcal{M}_t^L} \left| \left[ \mathbb{E} [G(\pi, m)] - \min_{m = \{m'_t\}_{t'=t}^{t+1}} \mathbb{E} [G(\pi, m)] \right] \right| \tag{32}$$

where first equality is the definition of the robust objective, second equality is decomposition of minimum across time steps and final inequality is simply a property of the min such as $|\min a - \min b| \leq \sup |a - b|$.

Finally plugging 28 in (32), it holds that

$$\left| J^R \left( \pi, t \right) - J^R \left( \pi, t+1 \right) \right| \tag{33}$$

$$= \left| \min_{m=\{m'_t\}^t_{t'=t_0} \in \mathcal{M}^L} \mathbb{E}_{\pi,P} \left[ G \left( \pi, m \right) \right] - \min_{m=\{m'_t\}^{t+1}_{t'=t_0} \in \mathcal{M}^L} \mathbb{E}_{\pi,P} \left[ G \left( \pi, m \right) \right] \right| \leq \frac{L_r}{1-\gamma} + \frac{\gamma L_P}{(1-\gamma)^2}. \tag{34}$$

$$:= L'. \tag{35}$$

Combining $t$ times the previous equation gives the result:

$$\forall t \in \mathbb{N}^+, \forall t_0 \in \mathbb{N}^+, \quad |J^R(\pi, t_0) - J^R(\pi, t_0 + t)| \leq L' t.$$

with $L' := \left( \frac{\gamma}{(1-\gamma)^2} L_P + \frac{1}{1-\gamma} L_r \right)$ $\qquad\square$

# E  Implementation details

## E.1  Algorithm

---
**Algorithm 2** Time-constrained robust training

---
**Input:** Time-constrained MDP: $(S, A, \Psi, p_\psi, r, L)$, Agent $\pi$, Adversary $\bar{\pi}$
**for** *each interaction time step $t$* **do**

$\quad a_t \sim \pi_t(s_t, \psi_t)$                 `// Sample an action with  Oracle-TC`
$\quad a_t \sim \pi_t(s_t, a_{t-1}, s_{t-1})$       `// Sample an action with  Stacked-TC`
$\quad a_t \sim \pi_t(s_t)$                     `// Sample an action with TC`
$\quad \psi_{t+1} \sim \bar{\pi}_\phi(s_t, a_t, \psi_t)$        `// Sample the worst TC parameter`
$\quad s_{t+1} \sim p_{\psi_{t+1}}(s_t, a_t)$           `// Sample a transition`
$\quad \mathcal{B} \leftarrow \mathcal{B} \cup \{(s_t, a_t, r(s_t, a_t), \psi_t, \psi_{t+1}, s_{t+1})\}$     `// Add transition to replay buffer`
$\quad \{s_i, a_i, r(s_i, a_i), \psi_i, \psi_{i+1}, s_{i+1}\}_{i \in [1,N]} \sim \mathcal{B}$    `// Sample a mini-batch of transitions`
$\quad \theta_c \leftarrow \theta_c - \alpha \nabla_{\theta_c} L_Q(\theta_c)$         `// Critic update phase`
$\quad \theta_a \leftarrow \theta_a - \alpha \nabla_{\theta_a} L_\pi(\theta_a)$              `// Actor update`
$\quad \phi_c \leftarrow \phi_c + \alpha \nabla_{\phi_c} L_{\bar{Q}}(\phi_c)$       `// Adversary Critic update phase`
$\quad \phi_a \leftarrow \phi_a + \alpha \nabla_{\phi_a} L_{\bar{\pi}}(\phi_a)$            `// Adversary update`

---

Note that in Time-constrained robust training Algorithm in section E.1, $L_Q$ and $L_\pi$ are as defined by [31] double critics and target network updates are omitted here for clarity

In Table 4, for the stack algorithm, $s_i$ is defined as $s_i \leftarrow s_i \cup s_{i-1} \cup a_{i-1}$ for `Stacked-TC`, and for the `Oracle-TC` version, $s_i \leftarrow s_i \cup \psi_i$.

| Loss Function | Equation |
|---|---|
| $L_{Q_{\theta_c}}$ (TC-RARL) | $\mathbb{E}\left[ Q_{\theta_c}(s_i, a_i) - r(s_i, a_i) + \gamma \min_{j=1,2} Q_{\theta_c}(s_{i+1}, \pi(s_{i+1})) \right]$ |
| $L_\pi(\theta_a)$ (TC-RARL) | $-\mathbb{E}\left[ Q_{\theta_c}(s_i, \pi_{\theta_a}(s_i)) \right]$ |
| $L_{\bar{\pi}}(\theta_a)$ (TC-RARL) | $\mathbb{E}\left[ \bar{Q}_{\theta_c}(s_i, a_i, \bar{\pi}(s_i, a_i), \psi_i) \right]$ |
| $L_{\bar{Q}}(\theta_c)$ (TC-RARL) | $\mathbb{E}\left[ \bar{Q}_{\theta_c}(s_i, a_i) - r(s_i, a_i) + \gamma \min_{j=1,2} \bar{Q}_{\theta_c}(s_{i+1}, \pi_{\theta_a}(s_{i+1}), \bar{\pi}_{\theta_a}(s_{i+1}, a_{i+1}, \psi_{i+1})) \right]$ |
| $L_{Q_{\theta_c}}$ Shared (TC-M2TD3) | $\mathbb{E}\left[ Q_{\theta_c}(s_i, a_i) - r(s_i, a_i) + \gamma \min_{j=1,2} Q_{\theta_c}(s_{i+1}, \pi_{\theta_a}(s_{i+1}), \bar{\pi}_{\theta_a}(s_{i+1}, a_{i+1}, \psi_{i+1})) \right]$ |
| $L_\pi(\theta_a)$ (TC-M2TD3) | $\mathbb{E}\left[ Q_{\theta_c}(s_i, a_i, \bar{\pi}_{\theta_a}(s_i, a_i), \psi_i) \right]$ |
| $L_{\bar{\pi}}(\theta_a)$ (TC-M2TD3) | $-\mathbb{E}\left[ \bar{Q}_{\theta_c}(s_i, a_i, \bar{\pi}_{\theta_a}(s_i, a_i, \psi_i)) \right]$ |

Table 4: Summary of Loss Functions for TD3 in TC-RARL and TC-M2TD3

## E.2 Neural network architecture

We employ a consistent neural network architecture for both the baseline and our proposed methods for the actor and the critic components. The architecture's design ensures uniformity and comparability across different models.

The critic network is structured with three layers, as depicted in Figure 3a, the critic begins with an input layer that takes the state and action as inputs, which then passes through two fully connected linear layers of 256 units each. The final layer is a single linear unit that outputs a real-valued function, representing the estimated value of the state-action pair.

The actor neural network, shown in Figure 3b, also utilizes a three-layer design. It begins with an input layer that accepts the state as input. This is followed by two linear layers, each consisting of 256 units. The output layer of the actor neural network has a dimensionality equal to the number of dimensions of the action space.

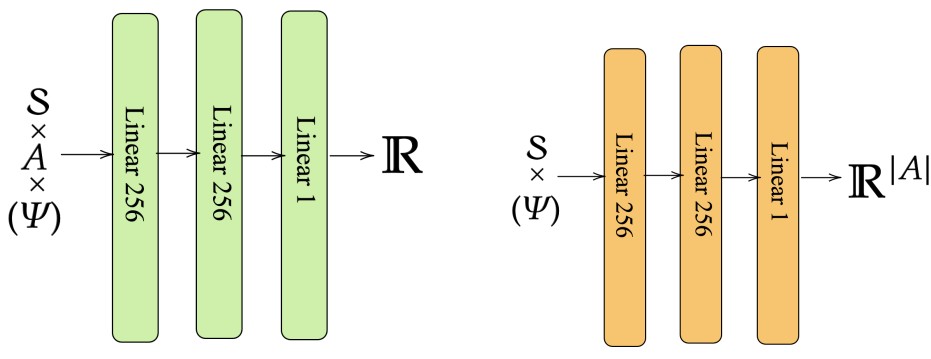

(a) Critic neural network architecture          (b) Actor neural network architecture

Figure 3: Actor critic neural network architecture

## E.3 M2TD3

We utilized the official M2TD3 [18] implementation provided by the original authors, accessible via the GitHub repository for M2TD3 and Oracle M2TD3.

For the TC-M2TD3 or variants, we implemented the M2TD3 algorithm as specified. To simplify our approach, we omitted the implementation of the multiple $\hat{\psi}$ network and the system for resetting $\hat{\psi}$. We replace with an adversary which $\bar{\pi} : \mathcal{S} \times \mathcal{A} \times \Psi \to \Psi$ which minimize $Q(s, a, \psi)$.

## E.4 TD3

We adopted the TD3 implementation from the CleanRL library, as detailed in [32].

## F Sanity check on the adversary training in the time-constrained evaluation

A natural question arises regarding the worst time-constrained perturbation. Whether we adequately trained the adversary in the loop, or its suboptimal performance might lead to overestimating the trained agent reward against the worst-case time-constrained perturbation. We monitored the adversary's performance during its training against a fixed agent to address this. The attached figure shows the episodic reward (from the agent's perspective) during the adversary's training over 5 million timesteps, with a perturbation radius of $L = 0.001$. Each curve is an average of over 10 seeds. The plots show a rapid decline in reward during the initial stages of training, followed by quick stabilization. The episodic reward stabilizes early in the Ant (Figure 4a) environment, indicating

| Hyperparameter | Default Value |
|---|---|
| Policy Std Rate | 0.1 |
| Policy Noise Rate | 0.2 |
| Noise Clip Policy Rate | 0.5 |
| Noise Clip Omega Rate | 0.5 |
| Omega Std Rate | 1.0 |
| Min Omega Std Rate | 0.1 |
| Maximum Steps | 5e6 |
| Batch Size | 100 |
| Hatomega Number | 5 |
| Replay Size | 1e6 |
| Policy Hidden Size | 256 |
| Critic Hidden Size | 256 |
| Policy Learning Rate | 3e-4 |
| Critic Learning Rate | 3e-4 |
| Policy Frequency | 2 |
| Gamma | 0.99 |
| Polyak | 5e-3 |
| Hatomega Parameter Distance | 0.1 |
| Minimum Probability | 5e-2 |
| Hatomega Learning Rate (ho_lr) | 3e-4 |
| Optimizer | Adam |

Table 5: Hyperparameters for the M2TD3 Agent

| Hyperparameter | Default Value |
|---|---|
| Maximum Steps | 5e6 |
| Buffer Size | $1 \times 10^6$ |
| Learning Rate | $3 \times 10^{-4}$ |
| Gamma | 0.99 |
| Tau | 0.005 |
| Policy Noise | 0.2 |
| Exploration Noise | 0.1 |
| Learning Starts | $2.5 \times 10^4$ |
| Policy Frequency | 2 |
| Batch Size | 256 |
| Noise Clip | 0.5 |
| Action Min | -1 |
| Action Max | 1 |
| Optimizer | Adam |

Table 6: Hyperparameters for the TD3 Agent

quick convergence. Similarly, in the HalfCheetah (Figure 4b) environment, the reward shows a sharp initial decline and stabilizes, suggesting effective training. For Hopper (Figure 4c), the reward decreases and then levels off, reflecting adversary convergence. Although the reward is more variable in the HumanoidStandup (Figure 4d) environment, it ultimately reaches a steady state, confirming adequate training. Finally, in the Walker environment, the reward pattern demonstrates a quick drop followed by stabilization, indicating convergence. These observations confirm that the adversaries were not undertrained. The rapid convergence to a stable performance across all environments ensures the accuracy of the worst time-constrained perturbations estimated during training.

# G   Uncertainty set in MuJoCo environments

The experiments of Section 5 follow the evaluation protocol proposed by [18] and based on MuJoCo environments [5]. These environments are designed with a 3D uncertainty sets. Table 7 lists all environments evaluated and their uncertainty sets. The uncertainty sets column defines the ranges of

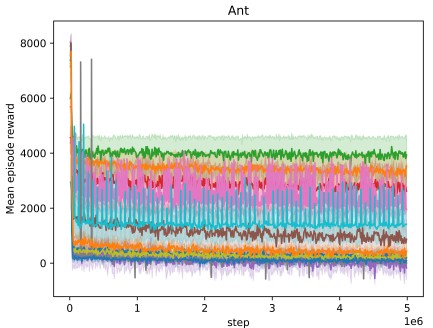

(a) Ant: Episodic reward of the trained agent during adversary training

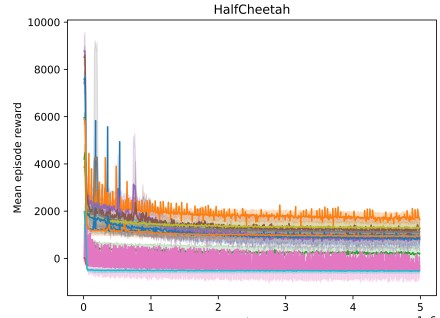

(b) HalfCheetah: Episodic reward of the trained agent during adversary training

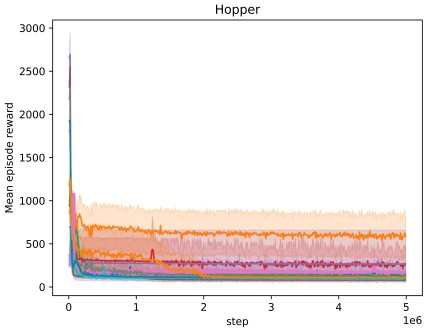

(c) Hopper: Episodic reward of the trained agent during adversary training

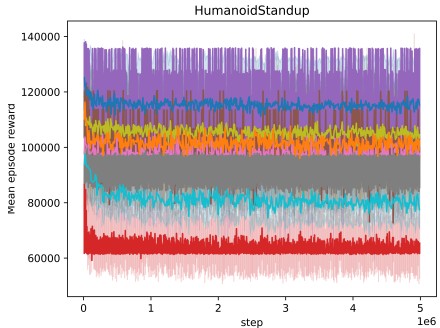

(d) HumanoidStandup: Episodic reward of the trained agent during adversary training

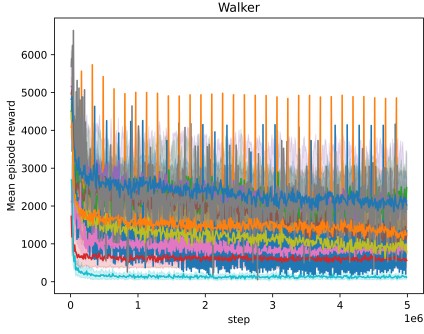

(e) Walker: Episodic reward of the trained agent during adversary training

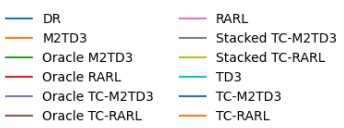

(f) Legend for algorithm

Figure 4: Episodic reward of the trained agent during the training of the adversary across different environments. Each plot represents the performance over 5 million timesteps, with rewards averaged across 10 seeds. The perturbation radius is set to $L = 0.001$ for all adversaries.

variation for the parameters within each environment. The reference parameters column indicates the nominal or default values. The uncertainty parameters column describes the physical meaning of each parameter.

# H Raw results

Table 8 reports the non-normalized time-constrained (with a radius of $L = 0.001$) worst-case scores, averaged across 10 independent runs for each benchmark. Table 9 reports the static worst case score obtained by each agent across a grid of environments, also averaged across 10 independent runs for

Table 7: List of environment and parameters for the experiments

| Environment | Uncertainty set $\mathcal{P}$ | Reference values | Uncertainty parameters |
|---|---|---|---|
| Ant | $[0.1, 3.0]$ $\times$ $[0.01, 3.0]$ $\times$ $[0.01, 3.0]$ | $(0.33, 0.04, 0.06)$ | torso mass; front left leg mass; front right leg mass |
| HalfCheetah | $[0.1, 4.0] \times [0.1, 7.0] \times [0.1, 3.0]$ | $(0.4, 6.36, 1.53)$ | world friction; torso mass; back thigh mass |
| Hopper | $[0.1, 3.0] \times [0.1, 3.0] \times [0.1, 4.0]$ | $(1.00, 3.53, 3.93)$ | world friction; torso mass; thigh mass |
| HumanoidStandup | $[0.1, 16.0]$ $\times$ $[0.1, 5.0] \times [0.1, 8.0]$ | $(8.32, 1.77, 4.53)$ | torso mass; right foot mass; left thigh mass |
| Walker | $[0.1, 4.0] \times [0.1, 5.0] \times [0.1, 6.0]$ | $(0.7, 3.53, 3.93)$ | world friction; torso mass; thigh mass |

Table 8: Avg. of time-constrained worst-case performance over 10 seeds for each method

| Environment Method | Ant | HalfCheetah | Hopper | HumanoidStandup | Walker |
|---|---|---|---|---|---|
| Oracle M2TD3 | $5768 \pm 395$ | $3521 \pm 187$ | $1241 \pm 125$ | $116232 \pm 1454$ | $4559 \pm 757$ |
| Oracle RARL | $4387 \pm 667$ | $-50 \pm 99$ | $344 \pm 113$ | $68979 \pm 10641$ | $1811 \pm 342$ |
| Oracle TC-M2TD3 | $7268 \pm 704$ | $7507 \pm 284$ | $\mathbf{3386 \pm 323}$ | $114411 \pm 16973$ | $5344 \pm 536$ |
| Oracle TC-RARL | $\mathbf{7534 \pm 781}$ | $\mathbf{7526 \pm 311}$ | $3169 \pm 311$ | $101182 \pm 12083$ | $4783 \pm 382$ |
| Stacked TC-M2TD3 | $6502 \pm 450$ | $6377 \pm 517$ | $3047 \pm 394$ | $85524 \pm 11448$ | $\mathbf{5724 \pm 828}$ |
| Stacked TC-RARL | $6955 \pm 690$ | $5319 \pm 223$ | $1747 \pm 153$ | $107913 \pm 5514$ | $4152 \pm 483$ |
| TC-M2TD3 | $7181 \pm 591$ | $6516 \pm 232$ | $2511 \pm 45$ | $\mathbf{129183 \pm 9120}$ | $4964 \pm 531$ |
| TC-RARL | $7473 \pm 361$ | $4989 \pm 284$ | $1475 \pm 158$ | $108669 \pm 17764$ | $3971 \pm 351$ |
| DR | $7247 \pm 925$ | $4986 \pm 363$ | $1642 \pm 104$ | $109618 \pm 11479$ | $4380 \pm 488$ |
| M2TD3 | $5622 \pm 435$ | $3671 \pm 405$ | $1120 \pm 220$ | $102839 \pm 12987$ | $4078 \pm 644$ |
| RARL | $4348 \pm 574$ | $382 \pm 366$ | $240 \pm 104$ | $106768 \pm 4051$ | $2388 \pm 559$ |
| TD3 | $2259 \pm 424$ | $1808 \pm 503$ | $777 \pm 407$ | $104877 \pm 12063$ | $1893 \pm 361$ |

each benchmark. Table 10 reports the static average case score obtained by each agent across a grid of environments, also averaged across 10 independent runs for each benchmark.

## H.1 Fixed adversary evaluation

At the beginning of each episode, $\psi_0 \sim \mathcal{U}(\Psi)$ is selected for every fixed adversary. The episode length is 1000 steps. To begin with, the random fixed adversary simulates stochastic changes. It selects a parameter $\psi_t$ at each timestep within a radius of $L = 0.1$, which is 100 times larger than in our training methods. This tests the agents' adaptability to unexpected changes. In contrast, the cosine fixed adversary introduces deterministic changes using a cosine function. The radius

Table 9: Avg. of raw static worst-case performance over 10 seeds for each method

| | Ant | HalfCheetah | Hopper | Humanoid | Walker |
|---|---|---|---|---|---|
| dr | $19.78 \pm 394.84$ | $2211.48 \pm 915.64$ | $245.01 \pm 167.21$ | $64886.87 \pm 30048.79$ | $1318.36 \pm 777.51$ |
| m2td3 | $2322.73 \pm 649.3$ | $2031.9 \pm 409.7$ | $273.6 \pm 131.9$ | $71900.97 \pm 24317.35$ | $2214.16 \pm 1330.4$ |
| oracle m2td3 | $2370.93 \pm 473.56$ | $319.67 \pm 599.26$ | $267.41 \pm 111.47$ | $93123.84 \pm 26696.17$ | $736.59 \pm 944.76$ |
| oracle rarl | $1396.88 \pm 777.46$ | $-278.84 \pm 54.36$ | $167.5 \pm 38.2$ | $45635.24 \pm 15974.44$ | $459.74 \pm 437.02$ |
| oracle tc m2td3 | $120.74 \pm 618.23$ | $4273.31 \pm 246.91$ | $168.7 \pm 217.94$ | $58687.26 \pm 22321.77$ | $710.99 \pm 799.08$ |
| oracle tc rarl | $1328.27 \pm 890.49$ | $3458.52 \pm 893.22$ | $150.54 \pm 33.12$ | $73276.78 \pm 9110.33$ | $1299.88 \pm 812.63$ |
| rarl | $960.11 \pm 744.01$ | $-211.8 \pm 218.73$ | $170.46 \pm 45.73$ | $67821.86 \pm 21555.24$ | $360.31 \pm 186.06$ |
| stacked tc m2td3 | $-242.98 \pm 212.98$ | $3467.34 \pm 418.64$ | $289.37 \pm 182.18$ | $58515.04 \pm 19186.25$ | $2475.58 \pm 1057.03$ |
| stacked tc rarl | $37.77 \pm 320.71$ | $1414.37 \pm 876.91$ | $344.37 \pm 190.1$ | $77357.17 \pm 18186.34$ | $1518.86 \pm 668.13$ |
| td3 | $-123.64 \pm 824.35$ | $-546.21 \pm 158.81$ | $69.3 \pm 42.77$ | $64577.24 \pm 16606.51$ | $114.41 \pm 211.05$ |
| tc m2td3 | $-271.34 \pm 191.15$ | $3286.67 \pm 603.14$ | $333.36 \pm 60.04$ | $73428.2 \pm 17879.28$ | $2603.59 \pm 706.63$ |
| tc rarl | $209.04 \pm 575.89$ | $1738.59 \pm 782.71$ | $376.01 \pm 155.4$ | $74840.68 \pm 33496.45$ | $1513.65 \pm 1239.3$ |

Table 10: Avg. of raw static average case performance over 10 seeds for each method

| env name algo-name | Ant | HalfCheetah | Hopper | HumanoidStandup | Walker |
|---|---|---|---|---|---|
| dr | $7500.88 \pm 143.38$ | $6170.33 \pm 442.57$ | $1688.36 \pm 225.59$ | $110939.89 \pm 22396.41$ | $4611.24 \pm 463.42$ |
| m2td3 | $5577.41 \pm 316.95$ | $4000.98 \pm 314.76$ | $1193.32 \pm 254.9$ | $109598.43 \pm 12992.35$ | $4311.2 \pm 877.89$ |
| oracle m2td3 | $5958.21 \pm 237.32$ | $4930.18 \pm 390.96$ | $1249.62 \pm 212.74$ | $118273.54 \pm 13891.06$ | $4616.05 \pm 407.94$ |
| oracle rarl | $4684.83 \pm 648.14$ | $36.19 \pm 216.52$ | $380.39 \pm 110.14$ | $76920.58 \pm 26135.3$ | $1451.39 \pm 1132.87$ |
| oracle-tc m2td3 | $7739.65 \pm 254.65$ | $9536.92 \pm 429.14$ | $3281.92 \pm 61.79$ | $119737.21 \pm 12697.2$ | $5442.85 \pm 499.78$ |
| oracle-tc-rarl | $7889.1 \pm 56.0$ | $9474.0 \pm 341.69$ | $3071.17 \pm 220.39$ | $104348.01 \pm 26249.98$ | $5220.2 \pm 318.07$ |
| rarl | $4650.55 \pm 395.03$ | $206.71 \pm 887.25$ | $276.37 \pm 52.42$ | $104764.87 \pm 17400.85$ | $2493.26 \pm 1113.74$ |
| stacked tc m2td3 | $6912.76 \pm 1116.81$ | $8583.55 \pm 479.97$ | $3124.06 \pm 133.27$ | $88039.74 \pm 15138.11$ | $5809.54 \pm 703.92$ |
| stacked-tc-rarl | $7123.07 \pm 332.33$ | $6130.71 \pm 384.05$ | $2072.75 \pm 306.48$ | $110843.2 \pm 19887.32$ | $4596.79 \pm 619.2$ |
| vanilla | $2600.43 \pm 1468.87$ | $2350.58 \pm 357.12$ | $733.18 \pm 382.06$ | $100533.0 \pm 12298.37$ | $2965.47 \pm 685.39$ |
| vanilla-tcm2td3 | $7366.9 \pm 169.58$ | $8467.64 \pm 397.42$ | $2756.5 \pm 273.91$ | $130305.38 \pm 22865.1$ | $5070.71 \pm 315.7$ |
| vanilla-tc-rarl | $7558.58 \pm 198.37$ | $6092.61 \pm 365.68$ | $1558.26 \pm 242.17$ | $108635.71 \pm 19848.21$ | $4325.42 \pm 283.04$ |

| Environment Method | Ant | HalfCheetah | Hopper | HumanoidStandup | Walker |
|---|---|---|---|---|---|
| Oracle TC-M2TD3 | $7782 \pm 915$ | $\mathbf{8805 \pm 165}$ | $\mathbf{2365 \pm 199}$ | $116791 \pm 12572$ | $5148 \pm 558$ |
| Oracle TC-RARL | $\mathbf{8041 \pm 470}$ | $8727 \pm 227$ | $2120 \pm 96$ | $107733 \pm 11975$ | $4896 \pm 326$ |
| Oracle M2TD3 | $5830 \pm 542$ | $4445 \pm 186$ | $1222 \pm 111$ | $118861 \pm 1365$ | $4584 \pm 787$ |
| Oracle RARL | $4628 \pm 514$ | $-51 \pm 60$ | $370 \pm 141$ | $81583 \pm 16526$ | $1829 \pm 356$ |
| Stacked TC-M2TD3 | $6888 \pm 738$ | $7400 \pm 385$ | $2114 \pm 138$ | $88436 \pm 10750$ | $\mathbf{5278 \pm 845}$ |
| Stacked TC-RARL | $7045 \pm 904$ | $5992 \pm 427$ | $1940 \pm 93$ | $106213 \pm 6770$ | $4430 \pm 389$ |
| TC-M2TD3 | $7156 \pm 692$ | $7530 \pm 185$ | $2157 \pm 112$ | $\mathbf{129599 \pm 13556}$ | $4931 \pm 568$ |
| TC-RARL | $7554 \pm 948$ | $5751 \pm 482$ | $1445 \pm 203$ | $105144 \pm 16813$ | $4112 \pm 329$ |
| DR | $7572 \pm 629$ | $6048 \pm 349$ | $1416 \pm 168$ | $105677 \pm 16333$ | $4371 \pm 431$ |
| M2TD3 | $5588 \pm 516$ | $4180 \pm 70$ | $1018 \pm 271$ | $107692 \pm 10414$ | $4176 \pm 783$ |
| RARL | $4347 \pm 567$ | $240 \pm 250$ | $390 \pm 130$ | $103583 \pm 9217$ | $1925 \pm 501$ |
| TD3 | $4017 \pm 518$ | $2028 \pm 1250$ | $1944 \pm 246$ | $91205 \pm 11350$ | $2860 \pm 419$ |

Table 11: Avg. performance against time-constrained fixed random adversary with a radius $L = 0.1$ over 10 seeds for each method

of $L = 0.1$ scales the frequency of the cosine function, ensuring smooth and periodic variations. Additionally, a phase shift at the start of each episode ensures different starting points. Meanwhile, the linear fixed adversary employs a linear function. The parameters change linearly from the initial value to either one of a vertex of the uncertainty set $\Psi$ over 1000 steps. Furthermore, the exponential fixed adversary uses an exponential function. Parameters change exponentially from the initial value to either of a vertex of the uncertainty set $\Psi$ over 1000 steps. This ensures smooth and predictable variations. Similarly, the logarithmic fixed adversary uses a logarithmic function. Parameters change logarithmically from the initial value to either of a vertex of the uncertainty of the uncertainty set $\Psi$ over 1000 steps, ensuring smooth and predictable variations. Agents trained under the time-constrained framework outperform all baselines in all environments for each fixed adversary, except when compared to the oracle TC method, which has access to $\psi$. In this case, the stacked-TC or TC methods outperform all baselines in all environments for the cosine, logarithmic, and exponential adversaries and outperform the fixed adversary baseline in 4 out of 5 instances for the random and linear fixed adversaries.

## H.2 Agents training curve

We conducted training for each agent over a duration of 5 million steps, closely monitoring the cumulative rewards obtained over a trajectory spanning 1,000 steps. To enhance the reliability of our results, we averaged the performance curves across 10 different seeds. The graphs in Figures 5 to 15 illustrate how different training methods, including Domain Randomization, M2TD3, RARL, Oracle RARL ,Oracle M2TD3, TC RARL, TC M2TD3, Stacked TC RARL and Stacked TC M2TD3, impact agent performance across various environments.

Table 12: Avg. performance against time-constrained fixed cosine adversary with a radius $L = 0.1$ over 10 seeds for each method

| Environment Method | Ant | HalfCheetah | Hopper | HumanoidStandup | Walker |
|---|---|---|---|---|---|
| Oracle M2TD3 | $5528 \pm 637$ | $3453 \pm 266$ | $1016 \pm 48$ | $119813 \pm 3281$ | $3589 \pm 863$ |
| Oracle RARL | $4550 \pm 626$ | $-79 \pm 34$ | $371 \pm 140$ | $74116 \pm 7890$ | $1593 \pm 326$ |
| Oracle TC-M2TD3 | $\mathbf{7586 \pm 1345}$ | $\mathbf{8174 \pm 383}$ | $\mathbf{1946 \pm 104}$ | $115506 \pm 12470$ | $4464 \pm 781$ |
| Oracle TC-RARL | $7522 \pm 1435$ | $7838 \pm 810$ | $1735 \pm 138$ | $110535 \pm 12702$ | $4442 \pm 591$ |
| Stacked TC-M2TD3 | $6269 \pm 849$ | $7173 \pm 509$ | $1734 \pm 157$ | $88157 \pm 10654$ | $\mathbf{4888 \pm 567}$ |
| Stacked TC-RARL | $6510 \pm 1395$ | $5385 \pm 445$ | $1519 \pm 118$ | $105696 \pm 5243$ | $3848 \pm 404$ |
| TC-M2TD3 | $6350 \pm 769$ | $6797 \pm 609$ | $1413 \pm 167$ | $\mathbf{130892 \pm 11544}$ | $4611 \pm 632$ |
| TC-RARL | $7124 \pm 912$ | $5109 \pm 348$ | $1172 \pm 129$ | $102864 \pm 13308$ | $3548 \pm 545$ |
| DR | $6975 \pm 992$ | $5490 \pm 384$ | $1091 \pm 169$ | $109227 \pm 17068$ | $3851 \pm 612$ |
| M2TD3 | $5330 \pm 684$ | $3634 \pm 321$ | $938 \pm 158$ | $108136 \pm 9755$ | $4126 \pm 644$ |
| RARL | $4153 \pm 602$ | $154 \pm 261$ | $363 \pm 58$ | $103366 \pm 7604$ | $1689 \pm 465$ |
| TD3 | $4025 \pm 557$ | $2784 \pm 370$ | $1317 \pm 189$ | $94352 \pm 10101$ | $2020 \pm 355$ |

Table 13: Avg. performance against a fixed linear adversary over 10 seeds for each method

| Environment Method | Ant | HalfCheetah | Hopper | HumanoidStandup | Walker |
|---|---|---|---|---|---|
| Oracle M2TD3 | $5811 \pm 121$ | $3560 \pm 167$ | $1216 \pm 326$ | $118829 \pm 846$ | $4431 \pm 615$ |
| Oracle RARL | $4447 \pm 600$ | $-122 \pm 64$ | $308 \pm 62$ | $81498 \pm 12860$ | $1503 \pm 450$ |
| Oracle TC-M2TD3 | $7919 \pm 595$ | $\mathbf{7495 \pm 268}$ | $\mathbf{2983 \pm 252}$ | $117610 \pm 11682$ | $4952 \pm 415$ |
| Oracle TC-RARL | $\mathbf{8069 \pm 151}$ | $7443 \pm 236$ | $2805 \pm 352$ | $110314 \pm 9354$ | $4613 \pm 257$ |
| Stacked TC-M2TD3 | $7003 \pm 812$ | $6365 \pm 335$ | $2714 \pm 198$ | $89556 \pm 11115$ | $\mathbf{5256 \pm 675}$ |
| Stacked TC-RARL | $7328 \pm 251$ | $5301 \pm 86$ | $1616 \pm 137$ | $105137 \pm 7903$ | $4234 \pm 385$ |
| TC-M2TD3 | $7622 \pm 413$ | $6451 \pm 246$ | $2228 \pm 131$ | $\mathbf{129501 \pm 10326}$ | $4844 \pm 417$ |
| TC-RARL | $7675 \pm 143$ | $4881 \pm 251$ | $1277 \pm 288$ | $105566 \pm 15551$ | $3906 \pm 381$ |
| DR | $7713 \pm 412$ | $5290 \pm 103$ | $1419 \pm 122$ | $108711 \pm 16696$ | $4307 \pm 309$ |
| M2TD3 | $5444 \pm 225$ | $3810 \pm 69$ | $970 \pm 323$ | $106311 \pm 9771$ | $4128 \pm 727$ |
| RARL | $4651 \pm 446$ | $218 \pm 138$ | $346 \pm 22$ | $101477 \pm 8947$ | $1894 \pm 515$ |
| TD3 | $3493 \pm 475$ | $1462 \pm 1246$ | $1722 \pm 366$ | $89934 \pm 10644$ | $2396 \pm 416$ |

Table 14: Avg. performance against a fixed logarithmic adversary over 10 seeds for each method

| Environment Method | Ant | HalfCheetah | Hopper | HumanoidStandup | Walker |
|---|---|---|---|---|---|
| Oracle M2TD3 | $5561 \pm 580$ | $3086 \pm 163$ | $957 \pm 165$ | $119214 \pm 2525$ | $4148 \pm 630$ |
| Oracle RARL | $4911 \pm 177$ | $-145 \pm 67$ | $293 \pm 49$ | $79522 \pm 13470$ | $1618 \pm 142$ |
| Oracle TC-M2TD3 | $7963 \pm 796$ | $\mathbf{6625 \pm 204}$ | $\mathbf{2577 \pm 171}$ | $116664 \pm 11798$ | $4818 \pm 451$ |
| Oracle TC-RARL | $\mathbf{8061 \pm 821}$ | $6532 \pm 304$ | $2572 \pm 177$ | $108213 \pm 10684$ | $4375 \pm 382$ |
| Stacked TC-M2TD3 | $7315 \pm 478$ | $5863 \pm 290$ | $2283 \pm 122$ | $87691 \pm 11133$ | $\mathbf{4931 \pm 735}$ |
| Stacked TC-RARL | $7514 \pm 62$ | $4770 \pm 145$ | $1426 \pm 197$ | $104193 \pm 8030$ | $3939 \pm 369$ |
| TC-M2TD3 | $7910 \pm 90$ | $5657 \pm 280$ | $1702 \pm 226$ | $\mathbf{128467 \pm 10762}$ | $4664 \pm 412$ |
| TC-RARL | $7686 \pm 208$ | $4475 \pm 238$ | $1082 \pm 298$ | $104835 \pm 16040$ | $3636 \pm 428$ |
| DR | $7883 \pm 67$ | $4721 \pm 146$ | $1166 \pm 332$ | $106171 \pm 16867$ | $3995 \pm 313$ |
| M2TD3 | $5371 \pm 279$ | $3565 \pm 105$ | $802 \pm 271$ | $104002 \pm 11606$ | $4206 \pm 712$ |
| RARL | $4620 \pm 763$ | $231 \pm 110$ | $340 \pm 44$ | $102004 \pm 9925$ | $1919 \pm 499$ |
| TD3 | $3678 \pm 623$ | $576 \pm 983$ | $1389 \pm 327$ | $88952 \pm 11367$ | $1956 \pm 360$ |

Table 15: Avg. performance against a fixed exponential adversary over 10 seeds for each method

| Environment Method | Ant | HalfCheetah | Hopper | HumanoidStandup | Walker |
|---|---|---|---|---|---|
| Oracle M2TD3 | $5860 \pm 93$ | $3780 \pm 137$ | $1271 \pm 224$ | $119205 \pm 1217$ | $4767 \pm 815$ |
| Oracle RARL | $4585 \pm 674$ | $-88 \pm 79$ | $302 \pm 41$ | $82063 \pm 13274$ | $1611 \pm 342$ |
| Oracle TC-M2TD3 | $7491 \pm 624$ | $\mathbf{8256 \pm 269}$ | $2894 \pm 244$ | $118476 \pm 11683$ | $5161 \pm 289$ |
| Oracle TC-RARL | $\mathbf{7724 \pm 368}$ | $8000 \pm 250$ | $\mathbf{3036 \pm 293}$ | $110092 \pm 10754$ | $4650 \pm 503$ |
| Stacked TC-M2TD3 | $6903 \pm 365$ | $7041 \pm 302$ | $2721 \pm 214$ | $91077 \pm 11945$ | $\mathbf{5310 \pm 882}$ |
| Stacked TC-RARL | $7061 \pm 222$ | $5741 \pm 249$ | $1825 \pm 145$ | $104793 \pm 6758$ | $4376 \pm 342$ |
| TC-M2TD3 | $7318 \pm 299$ | $7139 \pm 387$ | $2408 \pm 113$ | $\mathbf{129966 \pm 10823}$ | $4910 \pm 663$ |
| TC-RARL | $7441 \pm 133$ | $5326 \pm 220$ | $1457 \pm 163$ | $106491 \pm 14605$ | $4017 \pm 439$ |
| DR | $7389 \pm 206$ | $5691 \pm 121$ | $1564 \pm 99$ | $106290 \pm 17502$ | $4224 \pm 660$ |
| M2TD3 | $5466 \pm 318$ | $3909 \pm 332$ | $1062 \pm 272$ | $107097 \pm 9551$ | $4274 \pm 582$ |
| RARL | $4556 \pm 729$ | $228 \pm 181$ | $351 \pm 24$ | $102096 \pm 8291$ | $2053 \pm 493$ |
| TD3 | $3771 \pm 228$ | $2302 \pm 343$ | $2201 \pm 219$ | $90496 \pm 9487$ | $2768 \pm 538$ |

Table 16: Average wall-clock time for each algorithm

|  | Wall-clock time |
|---|---|
| TD3 | 14h |
| M2TD3 | 16h |
| RARL | 18h |
| TC | 16h |
| Stacked TC | 16h |
| Oracle TC | 16h |

## I Computer ressources

All experiments were run on a desktop machine (Intel i9, 10th generation processor, 64GB RAM) with a single NVIDIA RTX 4090 GPU. Averages and standard deviations were computed from 10 independent repetitions of each experiment.

## J Broader impact

This paper aims to advance robust reinforcement learning. It addresses general mathematical and computational challenges. These challenges may have societal and technological impacts, but we do not find it necessary to highlight them here.

### J.1 Limitations

While our proposed Time-Constrained Robust Markov Decision Process (TC-RMDP) framework significantly advances robust reinforcement learning by addressing multifactorial, correlated, and time-dependent disturbances, several limitations must be acknowledged. The TC-RMDP framework assumes that the parameter vector $\psi$ that governs environmental disturbances is known during training. In real-world applications, obtaining such detailed information may not always be feasible. This reliance on precise parameter knowledge limits the practical deployment of our algorithms in environments where $\psi$ cannot be accurately measured or inferred. Our approach assumes that the environment's dynamics can be accurately parameterized and that these parameters remain within a predefined uncertainty set $\Psi$. This assumption might not hold in more complex or highly dynamic environments where disturbances are not easily parameterized or when the uncertainty set $\Psi$ cannot comprehensively capture all possible variations. Consequently, the robustness of the learned policies might degrade when facing disturbances outside the considered parameter space. Addressing these limitations in future work.

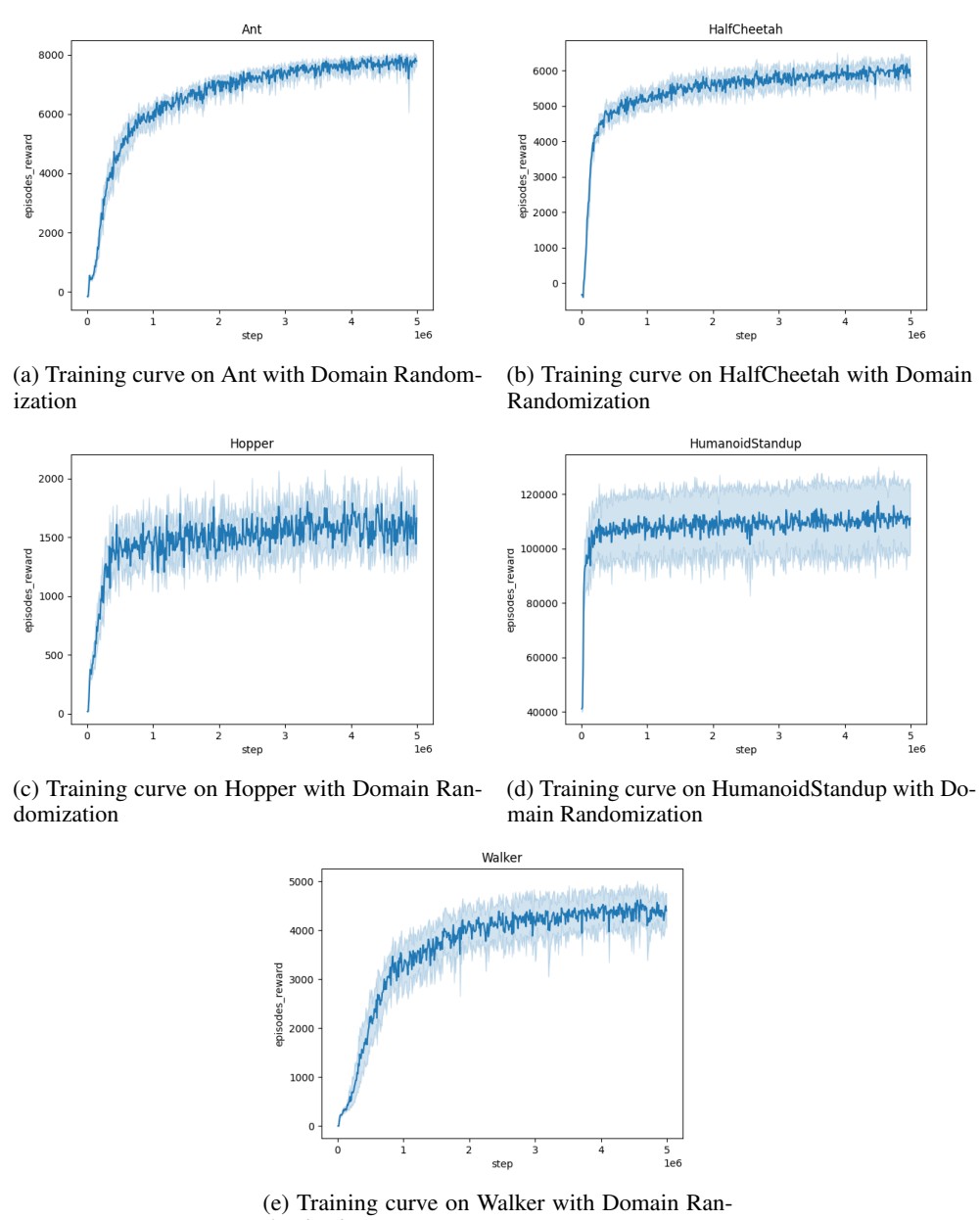

(a) Training curve on Ant with Domain Randomization

(b) Training curve on HalfCheetah with Domain Randomization

(c) Training curve on Hopper with Domain Randomization

(d) Training curve on HumanoidStandup with Domain Randomization

(e) Training curve on Walker with Domain Randomization

Figure 5: Averaged training curves for the Domain Randomization method over 10 seeds

## K  A heuristic for choosing the radius $L$

We used a simple heuristic to select the radius $L$ in our experiments. First, we normalized the parameters of the uncertainty set, scaling them to lie within the range $[0, 1]$ for each dimension. We then divided this range by the number of steps in an episode, denoted as $ep$, yielding $L = \frac{1}{ep}$. Our experiments, with $ep = 1000$, resulted in a radius of $L = 0.001$. This choice ensures that any initial parameter $\psi_0$ can move to any point within the uncertainty set's parameters for an episode.

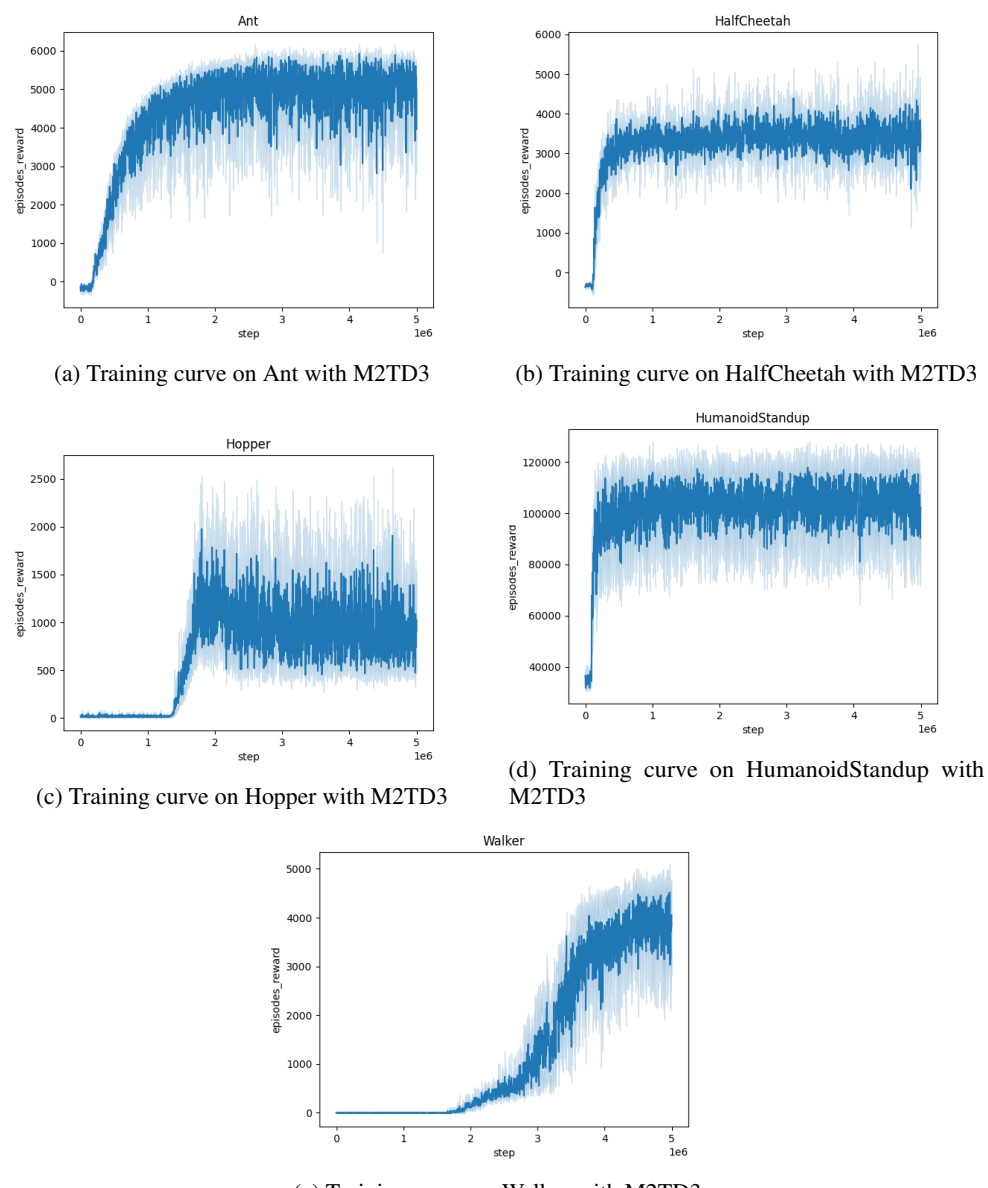

(a) Training curve on Ant with M2TD3

(b) Training curve on HalfCheetah with M2TD3

(c) Training curve on Hopper with M2TD3

(d) Training curve on HumanoidStandup with M2TD3

(e) Training curve on Walker with M2TD3

Figure 6: Averaged training curves for the M2TD3 method over 10 seeds


Figure 7: Averaged training curves for the RARL method over 10 seeds

- The answer NA means that the abstract and introduction do not include the claims made in the paper.
- The abstract and/or introduction should clearly state the claims made, including the contributions made in the paper and important assumptions and limitations. A No or NA answer to this question will not be perceived well by the reviewers.
- The claims made should match theoretical and experimental results, and reflect how much the results can be expected to generalize to other settings.
- It is fine to include aspirational goals as motivation as long as it is clear that these goals are not attained by the paper.

2. **Limitations**

   Question: Does the paper discuss the limitations of the work performed by the authors?

   Answer: [Yes]

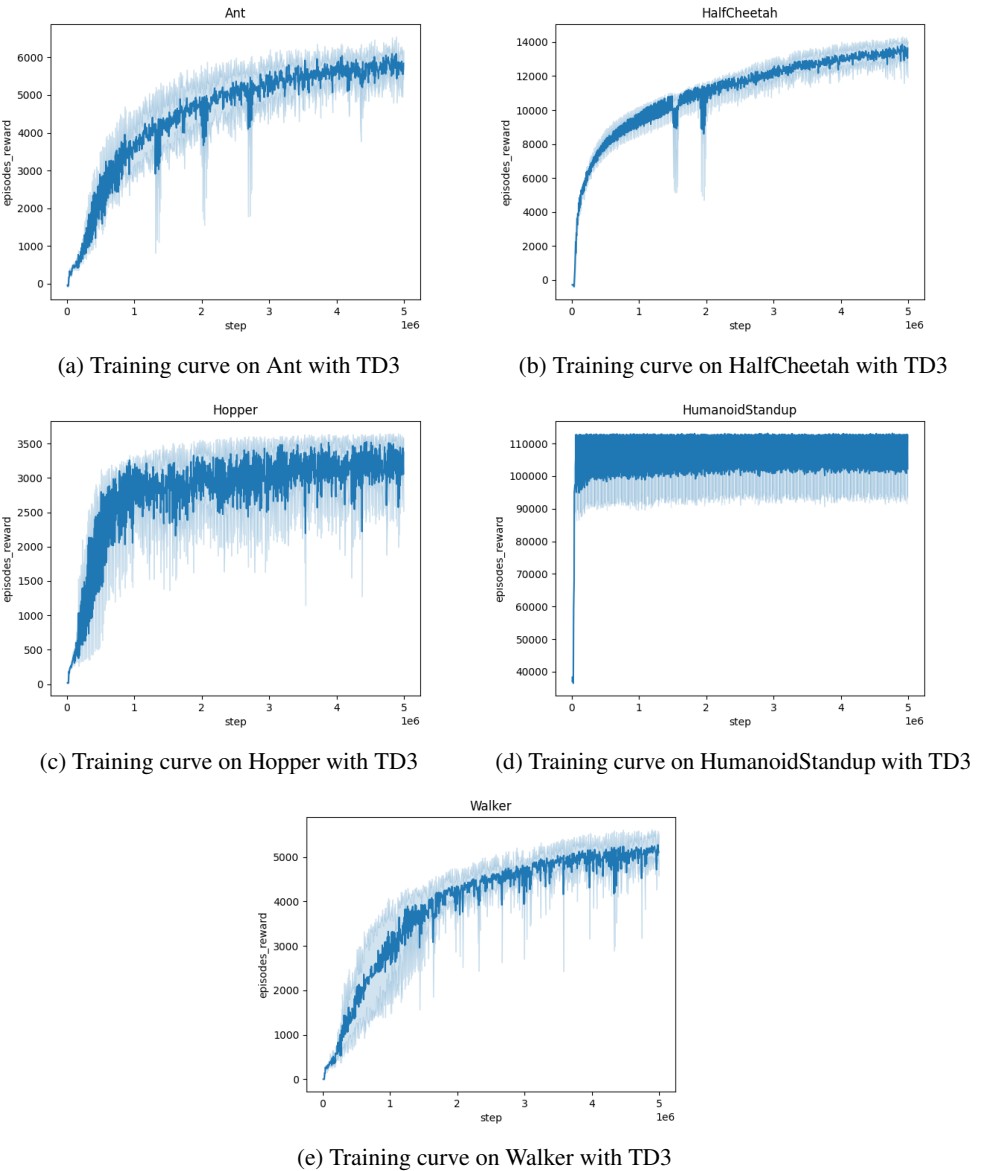

(a) Training curve on Ant with TD3

(b) Training curve on HalfCheetah with TD3

(c) Training curve on Hopper with TD3

(d) Training curve on HumanoidStandup with TD3

(e) Training curve on Walker with TD3

Figure 8: Averaged training curves for the TD3 method over 10 seeds

Justification: We have a section in the appendix discussing on the limitations of our work.

Guidelines:

- The answer NA means that the paper has no limitation while the answer No means that the paper has limitations, but those are not discussed in the paper.
- The authors are encouraged to create a separate "Limitations" section in their paper.
- The paper should point out any strong assumptions and how robust the results are to violations of these assumptions (e.g., independence assumptions, noiseless settings, model well-specification, asymptotic approximations only holding locally). The authors should reflect on how these assumptions might be violated in practice and what the implications would be.
- The authors should reflect on the scope of the claims made, e.g., if the approach was only tested on a few datasets or with a few runs. In general, empirical results often depend on implicit assumptions, which should be articulated.

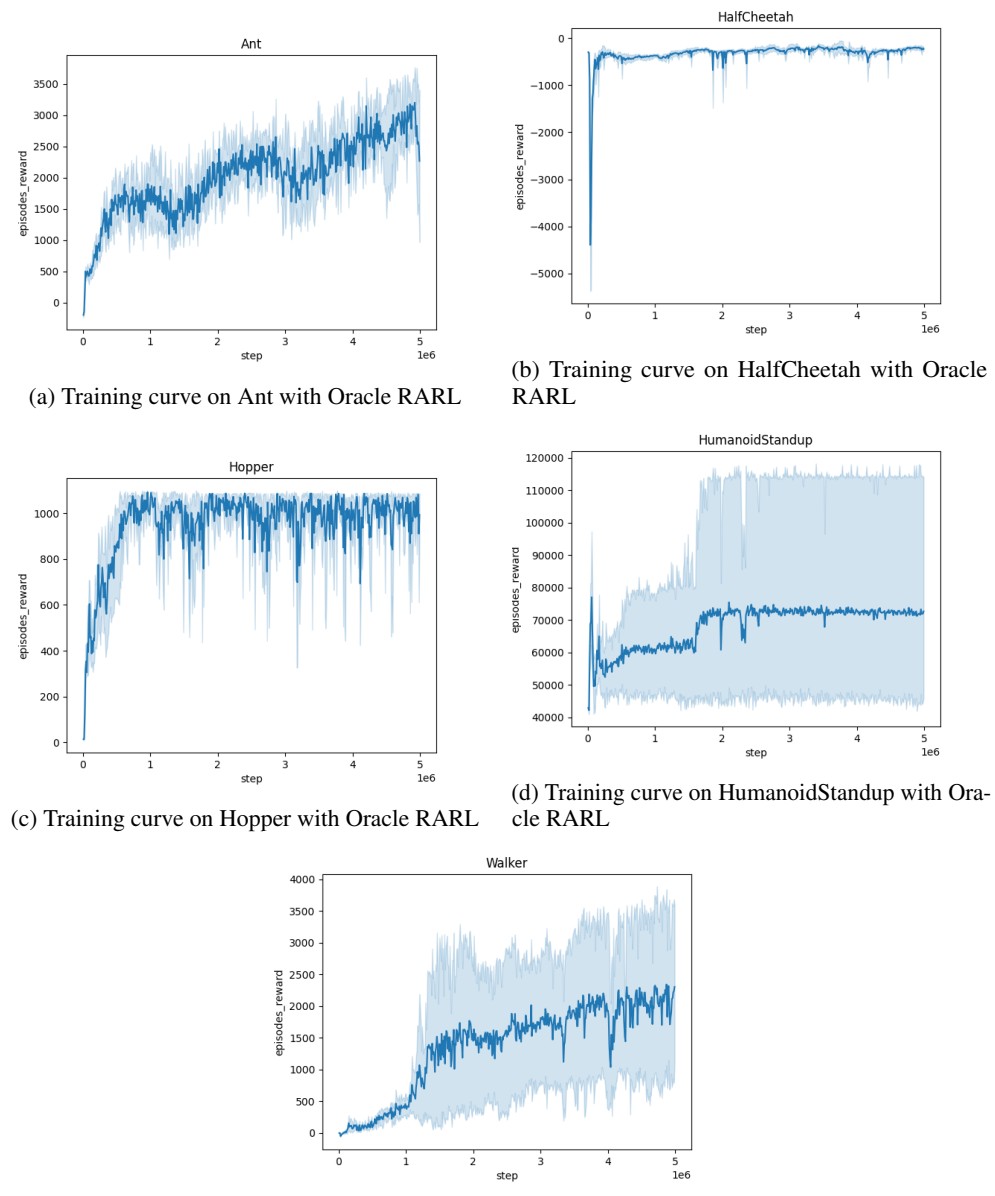

(a) Training curve on Ant with Oracle RARL

(b) Training curve on HalfCheetah with Oracle RARL

(c) Training curve on Hopper with Oracle RARL

(d) Training curve on HumanoidStandup with Oracle RARL

(e) Training curve on Walker with Oracle RARL

Figure 9: Averaged training curves for the Oracle RARL method over 10 seeds

- The authors should reflect on the factors that influence the performance of the approach. For example, a facial recognition algorithm may perform poorly when image resolution is low or images are taken in low lighting. Or a speech-to-text system might not be used reliably to provide closed captions for online lectures because it fails to handle technical jargon.

- The authors should discuss the computational efficiency of the proposed algorithms and how they scale with dataset size.

- If applicable, the authors should discuss possible limitations of their approach to address problems of privacy and fairness.

- While the authors might fear that complete honesty about limitations might be used by reviewers as grounds for rejection, a worse outcome might be that reviewers discover limitations that aren't acknowledged in the paper. The authors should use their best

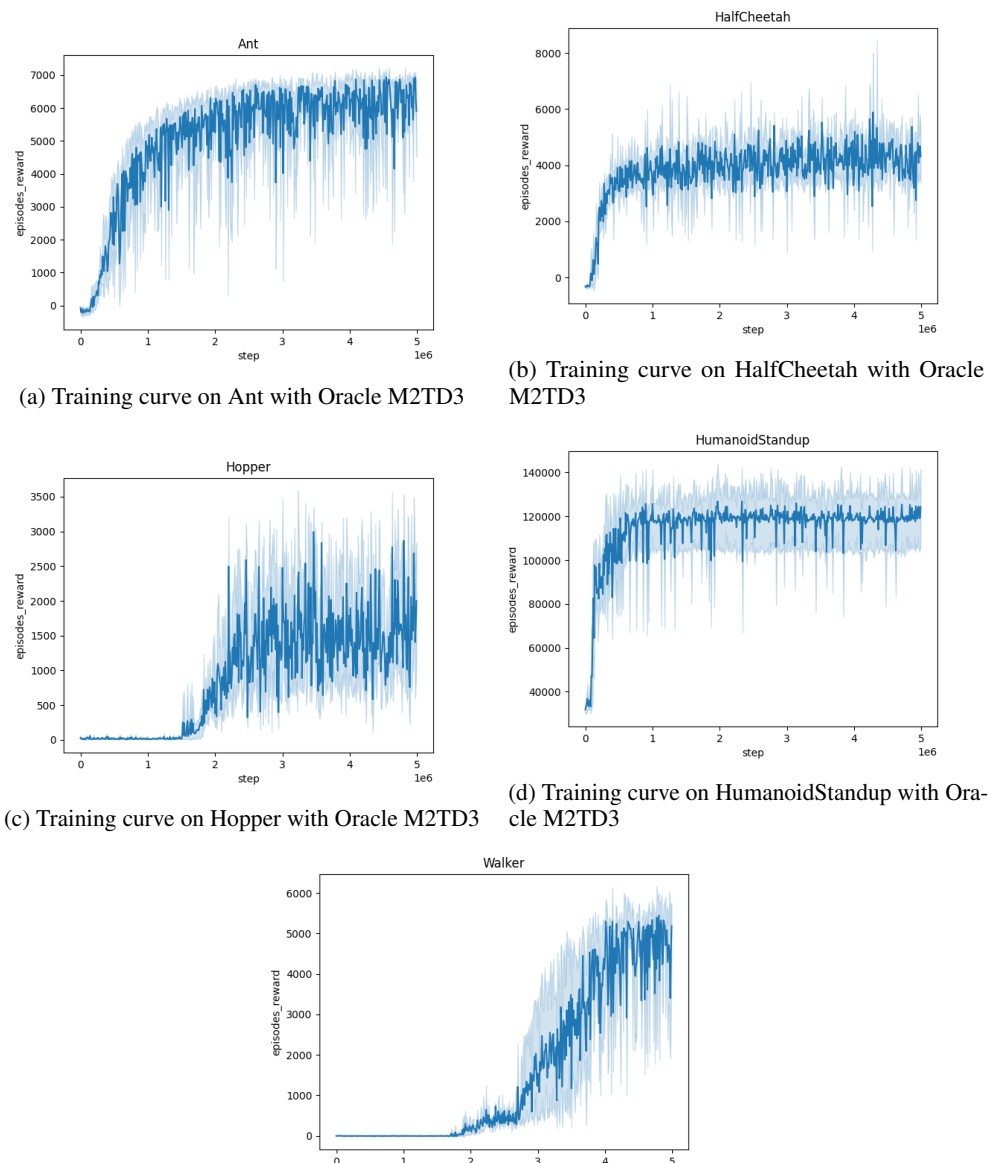

(a) Training curve on Ant with Oracle M2TD3

(b) Training curve on HalfCheetah with Oracle M2TD3

(c) Training curve on Hopper with Oracle M2TD3

(d) Training curve on HumanoidStandup with Oracle M2TD3

(e) Training curve on Walker with Oracle M2TD3

Figure 10: Averaged training curves for the Oracle M2TD3 method over 10 seeds

judgment and recognize that individual actions in favor of transparency play an important role in developing norms that preserve the integrity of the community. Reviewers will be specifically instructed to not penalize honesty concerning limitations.

3. **Theory Assumptions and Proofs**

Question: For each theoretical result, does the paper provide the full set of assumptions and a complete (and correct) proof?

Answer: [Yes]

Justification: Hypotheis of proof are well stated in Th.6.2 and 2.1

Guidelines:

- The answer NA means that the paper does not include theoretical results.

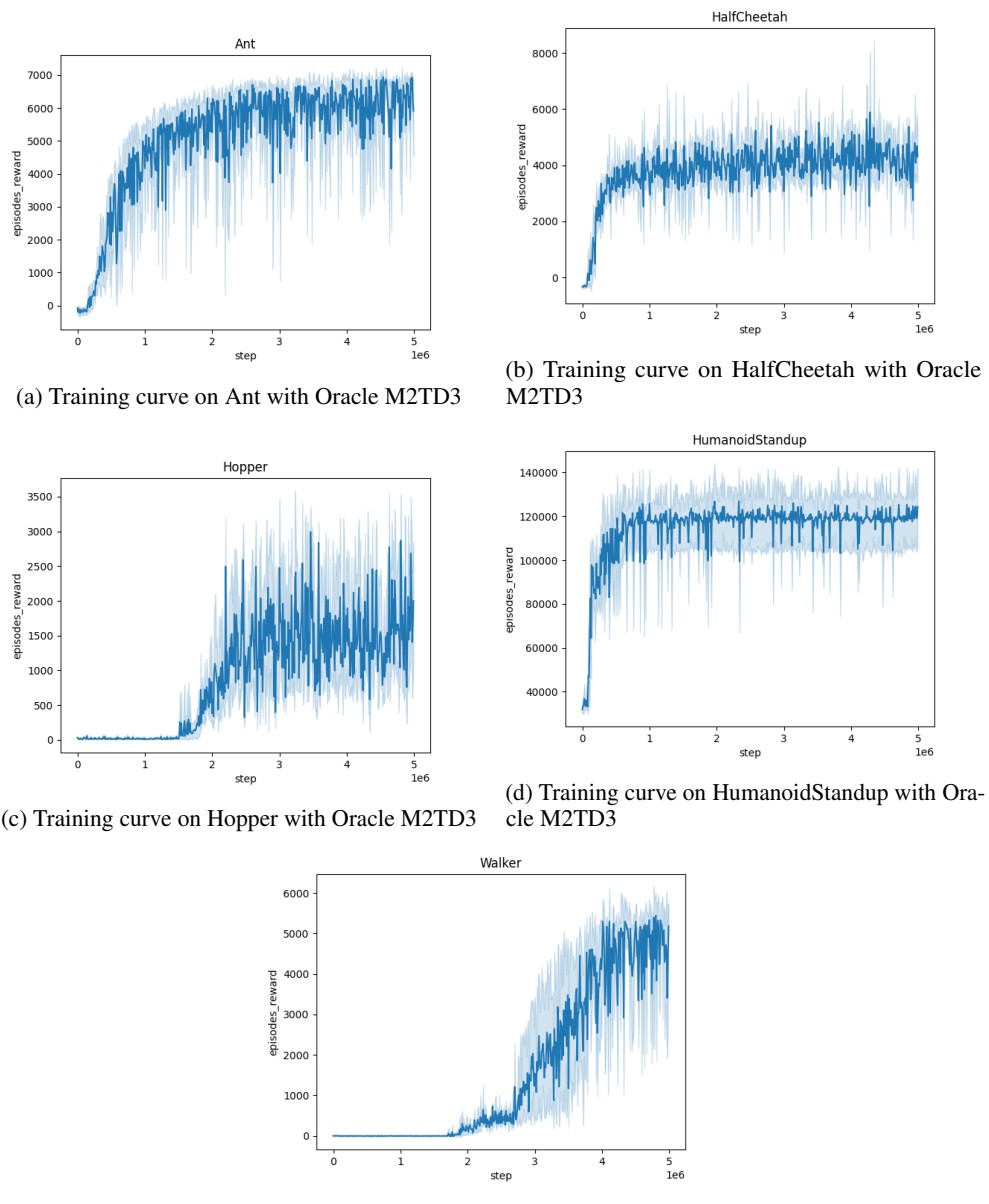

(a) Training curve on Ant with Oracle M2TD3

(b) Training curve on HalfCheetah with Oracle M2TD3

(c) Training curve on Hopper with Oracle M2TD3

(d) Training curve on HumanoidStandup with Oracle M2TD3

(e) Training curve on Walker with Oracle M2TD3

Figure 11: Averaged training curves for the Oracle M2TD3 method over 10 seeds

- All the theorems, formulas, and proofs in the paper should be numbered and cross-referenced.

- All assumptions should be clearly stated or referenced in the statement of any theorems.

- The proofs can either appear in the main paper or the supplemental material, but if they appear in the supplemental material, the authors are encouraged to provide a short proof sketch to provide intuition.

- Inversely, any informal proof provided in the core of the paper should be complemented by formal proofs provided in appendix or supplemental material.

- Theorems and Lemmas that the proof relies upon should be properly referenced.

4. **Experimental Result Reproducibility**

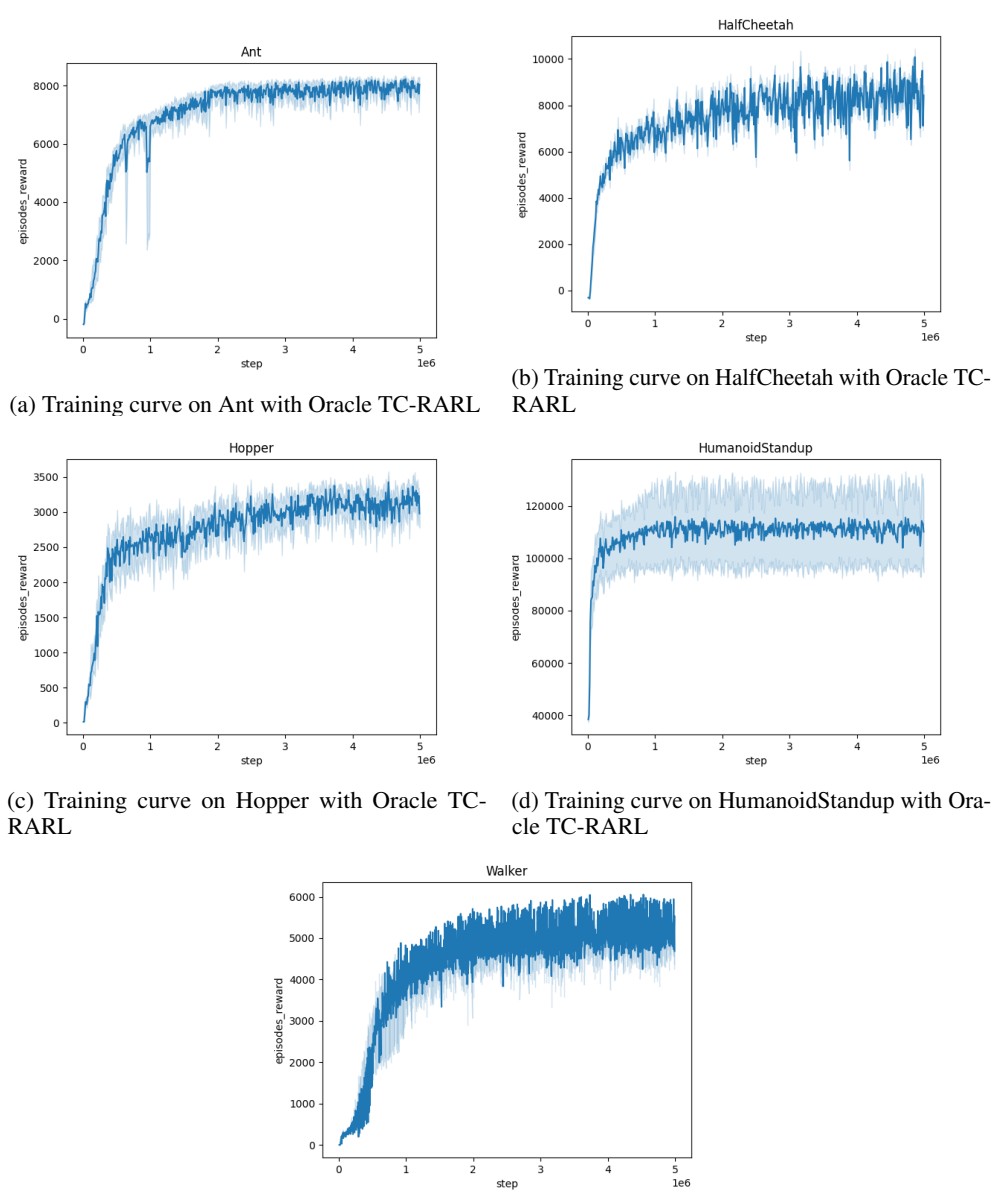

(a) Training curve on Ant with Oracle TC-RARL

(b) Training curve on HalfCheetah with Oracle TC-RARL

(c) Training curve on Hopper with Oracle TC-RARL

(d) Training curve on HumanoidStandup with Oracle TC-RARL

(e) Training curve on Walker with Oracle TC-RARL

Figure 12: Averaged training curves for the Oracle TC-RARL method over 10 seeds

Question: Does the paper fully disclose all the information needed to reproduce the main experimental results of the paper to the extent that it affects the main claims and/or conclusions of the paper (regardless of whether the code and data are provided or not)?

Answer: [Yes]

Justification: We provide the algorithm, all detail implementations, hyperparameters.

Guidelines:

- The answer NA means that the paper does not include experiments.
- If the paper includes experiments, a No answer to this question will not be perceived well by the reviewers: Making the paper reproducible is important, regardless of whether the code and data are provided or not.

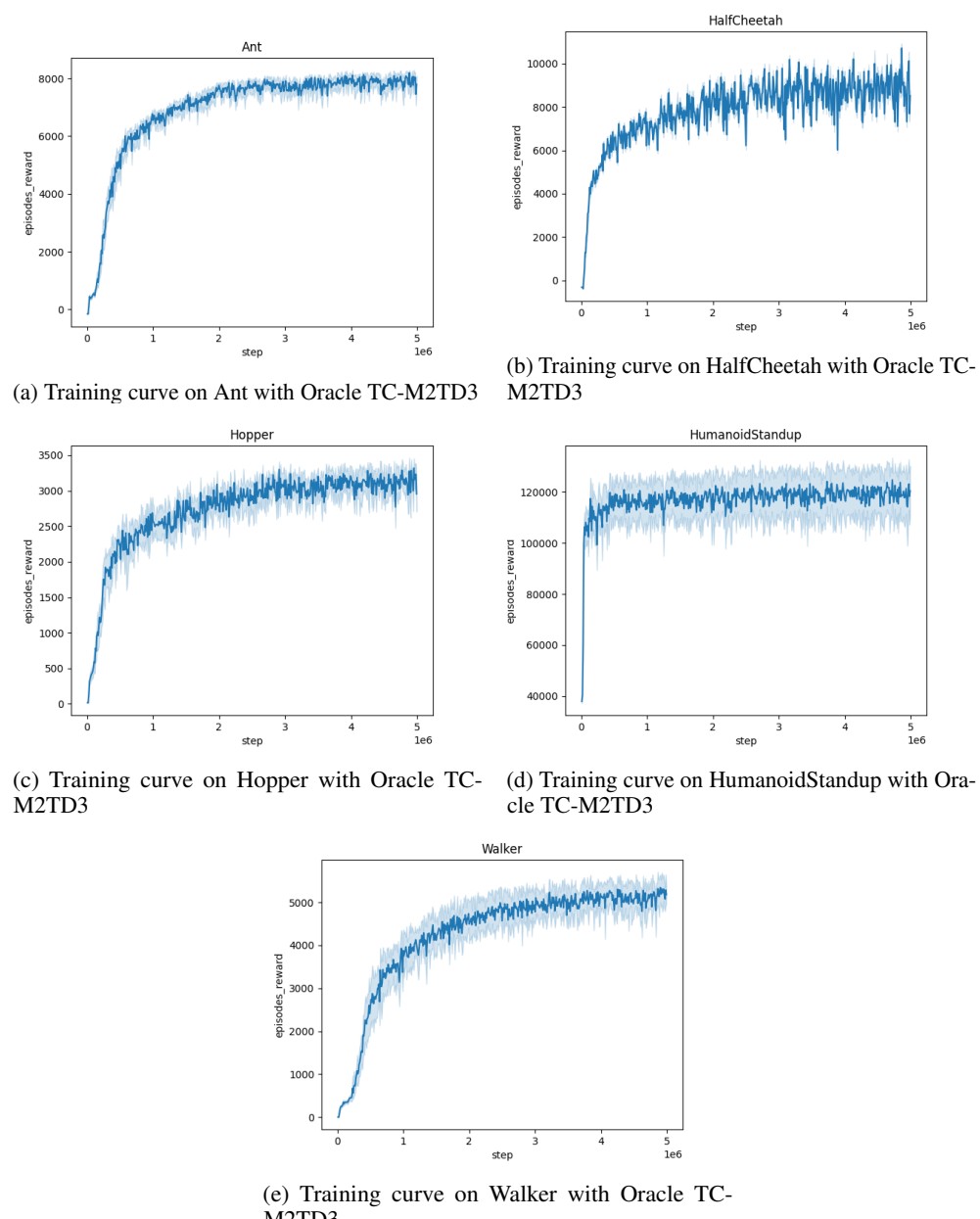

(a) Training curve on Ant with Oracle TC-M2TD3

(b) Training curve on HalfCheetah with Oracle TC-M2TD3

(c) Training curve on Hopper with Oracle TC-M2TD3

(d) Training curve on HumanoidStandup with Oracle TC-M2TD3

(e) Training curve on Walker with Oracle TC-M2TD3

Figure 13: Averaged training curves for the Oracle TC-M2TD3 method over 10 seeds

- If the contribution is a dataset and/or model, the authors should describe the steps taken to make their results reproducible or verifiable.

- Depending on the contribution, reproducibility can be accomplished in various ways. For example, if the contribution is a novel architecture, describing the architecture fully might suffice, or if the contribution is a specific model and empirical evaluation, it may be necessary to either make it possible for others to replicate the model with the same dataset, or provide access to the model. In general. releasing code and data is often one good way to accomplish this, but reproducibility can also be provided via detailed instructions for how to replicate the results, access to a hosted model (e.g., in the case of a large language model), releasing of a model checkpoint, or other means that are appropriate to the research performed.

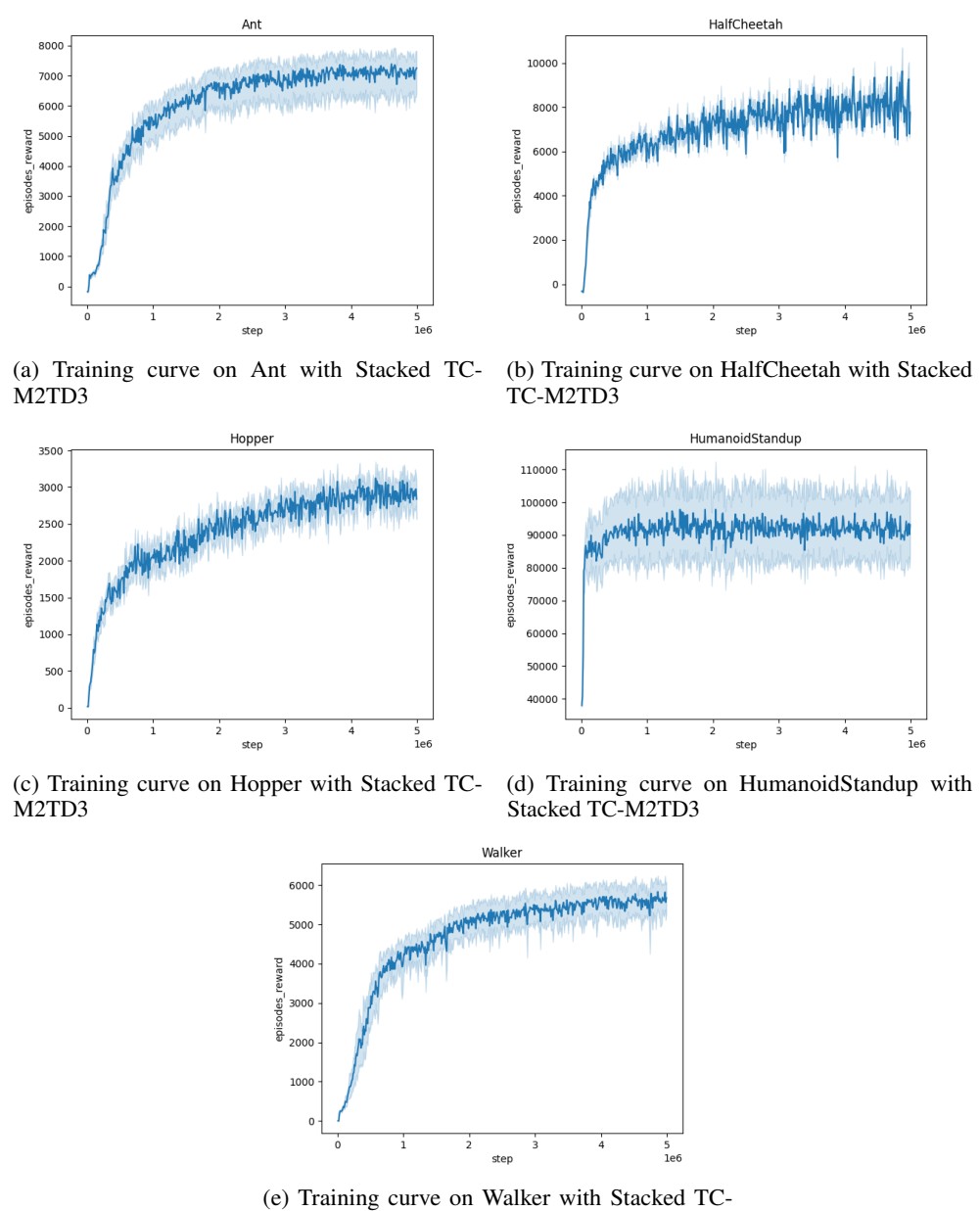

(a) Training curve on Ant with Stacked TC-M2TD3

(b) Training curve on HalfCheetah with Stacked TC-M2TD3

(c) Training curve on Hopper with Stacked TC-M2TD3

(d) Training curve on HumanoidStandup with Stacked TC-M2TD3

(e) Training curve on Walker with Stacked TC-M2TD3

Figure 14: Averaged training curves for the Stacked TC-M2TD3 method over 10 seeds

- While NeurIPS does not require releasing code, the conference does require all submissions to provide some reasonable avenue for reproducibility, which may depend on the nature of the contribution. For example

  (a) If the contribution is primarily a new algorithm, the paper should make it clear how to reproduce that algorithm.

  (b) If the contribution is primarily a new model architecture, the paper should describe the architecture clearly and fully.

  (c) If the contribution is a new model (e.g., a large language model), then there should either be a way to access this model for reproducing the results or a way to reproduce the model (e.g., with an open-source dataset or instructions for how to construct the dataset).

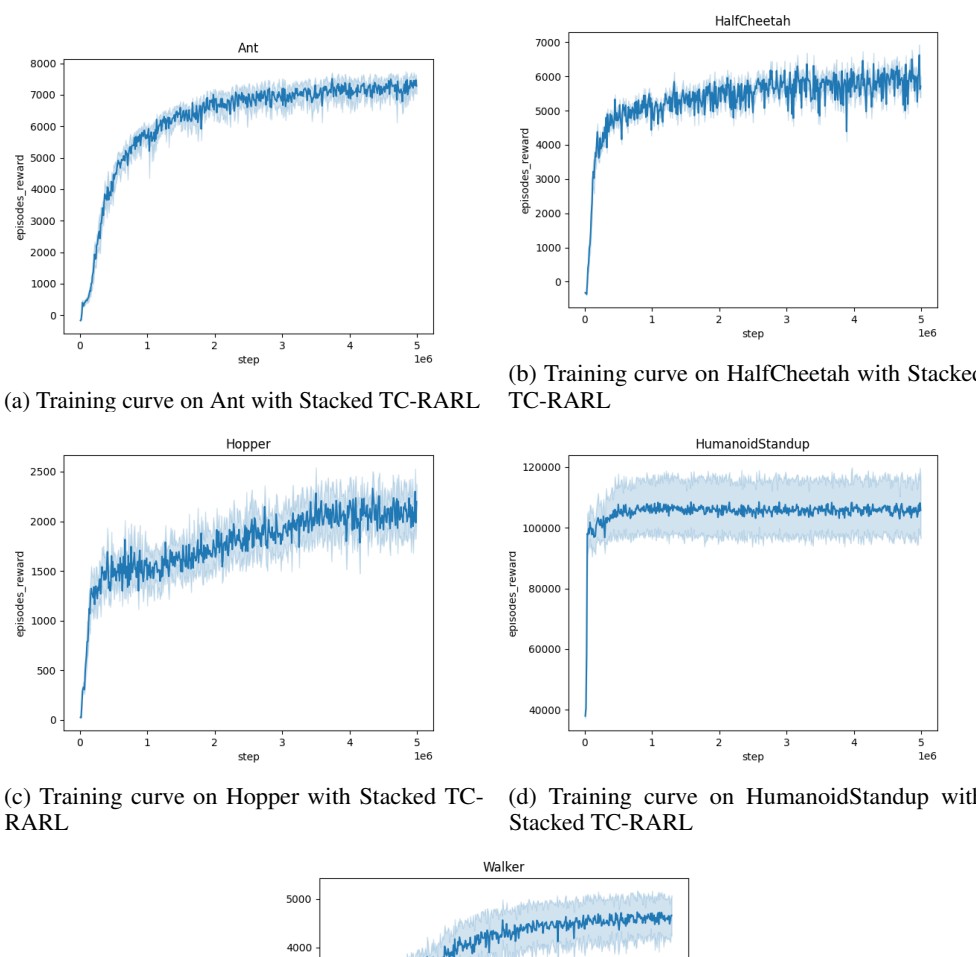

(a) Training curve on Ant with Stacked TC-RARL

(b) Training curve on HalfCheetah with Stacked TC-RARL

(c) Training curve on Hopper with Stacked TC-RARL

(d) Training curve on HumanoidStandup with Stacked TC-RARL

(e) Training curve on Walker with Stacked TC-RARL

Figure 15: Averaged training curves for the Stacked TC-RARL method over 10 seeds

(d) We recognize that reproducibility may be tricky in some cases, in which case authors are welcome to describe the particular way they provide for reproducibility. In the case of closed-source models, it may be that access to the model is limited in some way (e.g., to registered users), but it should be possible for other researchers to have some path to reproducing or verifying the results.

5. **Open access to data and code**

   Question: Does the paper provide open access to the data and code, with sufficient instructions to faithfully reproduce the main experimental results, as described in supplemental material?

   Answer: [Yes]

   Justification: We provide the code with clear instruction, to reproduce the experiments.

Guidelines:

- The answer NA means that paper does not include experiments requiring code.
- Please see the NeurIPS code and data submission guidelines (`https://nips.cc/public/guides/CodeSubmissionPolicy`) for more details.
- While we encourage the release of code and data, we understand that this might not be possible, so "No" is an acceptable answer. Papers cannot be rejected simply for not including code, unless this is central to the contribution (e.g., for a new open-source benchmark).
- The instructions should contain the exact command and environment needed to run to reproduce the results. See the NeurIPS code and data submission guidelines (`https://nips.cc/public/guides/CodeSubmissionPolicy`) for more details.
- The authors should provide instructions on data access and preparation, including how to access the raw data, preprocessed data, intermediate data, and generated data, etc.
- The authors should provide scripts to reproduce all experimental results for the new proposed method and baselines. If only a subset of experiments are reproducible, they should state which ones are omitted from the script and why.
- At submission time, to preserve anonymity, the authors should release anonymized versions (if applicable).
- Providing as much information as possible in supplemental material (appended to the paper) is recommended, but including URLs to data and code is permitted.

6. **Experimental Setting/Details**

   Question: Does the paper specify all the training and test details (e.g., data splits, hyper-parameters, how they were chosen, type of optimizer, etc.) necessary to understand the results?

   Answer: [Yes]

   Justification: We provides all hyperparameters, optimizer and reinforcement learning algorithm. All those details is provided in the appendix.

   Guidelines:

   - The answer NA means that the paper does not include experiments.
   - The experimental setting should be presented in the core of the paper to a level of detail that is necessary to appreciate the results and make sense of them.
   - The full details can be provided either with the code, in appendix, or as supplemental material.

7. **Experiment Statistical Significance**

   Question: Does the paper report error bars suitably and correctly defined or other appropriate information about the statistical significance of the experiments?

   Answer: [Yes]

   Justification: We profide our computers resources to runs all experiments.

   Guidelines:

   - The answer NA means that the paper does not include experiments.
   - The authors should answer "Yes" if the results are accompanied by error bars, confidence intervals, or statistical significance tests, at least for the experiments that support the main claims of the paper.
   - The factors of variability that the error bars are capturing should be clearly stated (for example, train/test split, initialization, random drawing of some parameter, or overall run with given experimental conditions).
   - The method for calculating the error bars should be explained (closed form formula, call to a library function, bootstrap, etc.)
   - The assumptions made should be given (e.g., Normally distributed errors).
   - It should be clear whether the error bar is the standard deviation or the standard error of the mean.

- It is OK to report 1-sigma error bars, but one should state it. The authors should preferably report a 2-sigma error bar than state that they have a 96% CI, if the hypothesis of Normality of errors is not verified.
- For asymmetric distributions, the authors should be careful not to show in tables or figures symmetric error bars that would yield results that are out of range (e.g. negative error rates).
- If error bars are reported in tables or plots, The authors should explain in the text how they were calculated and reference the corresponding figures or tables in the text.

8. **Experiments Compute Resources**

Question: For each experiment, does the paper provide sufficient information on the computer resources (type of compute workers, memory, time of execution) needed to reproduce the experiments?

Answer: [Yes]

Justification: We have a section in the appendix discussing of the computer resources for our work.

Guidelines:

- The answer NA means that the paper does not include experiments.
- The paper should indicate the type of compute workers CPU or GPU, internal cluster, or cloud provider, including relevant memory and storage.
- The paper should provide the amount of compute required for each of the individual experimental runs as well as estimate the total compute.
- The paper should disclose whether the full research project required more compute than the experiments reported in the paper (e.g., preliminary or failed experiments that didn't make it into the paper).

9. **Code Of Ethics**

Question: Does the research conducted in the paper conform, in every respect, with the NeurIPS Code of Ethics https://neurips.cc/public/EthicsGuidelines?

Answer: [Yes]

Justification: Our contribution aims to safer RL algorithms, and we conducted our experiment of open source benchmarks.

Guidelines:

- The answer NA means that the authors have not reviewed the NeurIPS Code of Ethics.
- If the authors answer No, they should explain the special circumstances that require a deviation from the Code of Ethics.
- The authors should make sure to preserve anonymity (e.g., if there is a special consideration due to laws or regulations in their jurisdiction).

10. **Broader Impacts**

Question: Does the paper discuss both potential positive societal impacts and negative societal impacts of the work performed?

Answer: [Yes]

Justification: We discuss on the broader impact in the conclusion and the appendix

Guidelines:

- The answer NA means that there is no societal impact of the work performed.
- If the authors answer NA or No, they should explain why their work has no societal impact or why the paper does not address societal impact.
- Examples of negative societal impacts include potential malicious or unintended uses (e.g., disinformation, generating fake profiles, surveillance), fairness considerations (e.g., deployment of technologies that could make decisions that unfairly impact specific groups), privacy considerations, and security considerations.

- The conference expects that many papers will be foundational research and not tied to particular applications, let alone deployments. However, if there is a direct path to any negative applications, the authors should point it out. For example, it is legitimate to point out that an improvement in the quality of generative models could be used to generate deepfakes for disinformation. On the other hand, it is not needed to point out that a generic algorithm for optimizing neural networks could enable people to train models that generate Deepfakes faster.
- The authors should consider possible harms that could arise when the technology is being used as intended and functioning correctly, harms that could arise when the technology is being used as intended but gives incorrect results, and harms following from (intentional or unintentional) misuse of the technology.
- If there are negative societal impacts, the authors could also discuss possible mitigation strategies (e.g., gated release of models, providing defenses in addition to attacks, mechanisms for monitoring misuse, mechanisms to monitor how a system learns from feedback over time, improving the efficiency and accessibility of ML).

11. **Safeguards**

Question: Does the paper describe safeguards that have been put in place for responsible release of data or models that have a high risk for misuse (e.g., pretrained language models, image generators, or scraped datasets)?

Answer: [NA]

Justification:

Guidelines:

- The answer NA means that the paper poses no such risks.
- Released models that have a high risk for misuse or dual-use should be released with necessary safeguards to allow for controlled use of the model, for example by requiring that users adhere to usage guidelines or restrictions to access the model or implementing safety filters.
- Datasets that have been scraped from the Internet could pose safety risks. The authors should describe how they avoided releasing unsafe images.
- We recognize that providing effective safeguards is challenging, and many papers do not require this, but we encourage authors to take this into account and make a best faith effort.

12. **Licenses for existing assets**

Question: Are the creators or original owners of assets (e.g., code, data, models), used in the paper, properly credited and are the license and terms of use explicitly mentioned and properly respected?

Answer: [Yes]

Justification: The authors cite the original paper that produced the code package or dataset.

Guidelines:

- The answer NA means that the paper does not use existing assets.
- The authors should cite the original paper that produced the code package or dataset.
- The authors should state which version of the asset is used and, if possible, include a URL.
- The name of the license (e.g., CC-BY 4.0) should be included for each asset.
- For scraped data from a particular source (e.g., website), the copyright and terms of service of that source should be provided.
- If assets are released, the license, copyright information, and terms of use in the package should be provided. For popular datasets, `paperswithcode.com/datasets` has curated licenses for some datasets. Their licensing guide can help determine the license of a dataset.
- For existing datasets that are re-packaged, both the original license and the license of the derived asset (if it has changed) should be provided.

- If this information is not available online, the authors are encouraged to reach out to the asset's creators.

13. **New Assets**

    Question: Are new assets introduced in the paper well documented and is the documentation provided alongside the assets?

    Answer: [NA]

    Justification:

    Guidelines:

    - The answer NA means that the paper does not release new assets.
    - Researchers should communicate the details of the dataset/code/model as part of their submissions via structured templates. This includes details about training, license, limitations, etc.
    - The paper should discuss whether and how consent was obtained from people whose asset is used.
    - At submission time, remember to anonymize your assets (if applicable). You can either create an anonymized URL or include an anonymized zip file.

14. **Crowdsourcing and Research with Human Subjects**

    Question: For crowdsourcing experiments and research with human subjects, does the paper include the full text of instructions given to participants and screenshots, if applicable, as well as details about compensation (if any)?

    Answer: [NA]

    Justification:

    Guidelines:

    - The answer NA means that the paper does not involve crowdsourcing nor research with human subjects.
    - Including this information in the supplemental material is fine, but if the main contribution of the paper involves human subjects, then as much detail as possible should be included in the main paper.
    - According to the NeurIPS Code of Ethics, workers involved in data collection, curation, or other labor should be paid at least the minimum wage in the country of the data collector.

15. **Institutional Review Board (IRB) Approvals or Equivalent for Research with Human Subjects**

    Question: Does the paper describe potential risks incurred by study participants, whether such risks were disclosed to the subjects, and whether Institutional Review Board (IRB) approvals (or an equivalent approval/review based on the requirements of your country or institution) were obtained?

    Answer: [NA]

    Justification:

    Guidelines:

    - The answer NA means that the paper does not involve crowdsourcing nor research with human subjects.
    - Depending on the country in which research is conducted, IRB approval (or equivalent) may be required for any human subjects research. If you obtained IRB approval, you should clearly state this in the paper.
    - We recognize that the procedures for this may vary significantly between institutions and locations, and we expect authors to adhere to the NeurIPS Code of Ethics and the guidelines for their institution.
    - For initial submissions, do not include any information that would break anonymity (if applicable), such as the institution conducting the review.

