# OpenReview forum: "Time-Constrained Robust MDPs"
_NeurIPS.cc/2024/Conference — NeurIPS 2024 poster_

### Official Review · Reviewer_aiQz · 2024-07-07

**Soundness:** 2
**Presentation:** 2
**Contribution:** 2
**Rating:** 5
**Confidence:** 2

**Summary:**

This paper proposed a novel concept, time-constrained robust MDP, to address the conservativeness issue of the rectangular assumption. They assume the transition depends on an underlying parameter, and the parameter can be adversarially chosen from an uncertainty set. Several algorithms are proposed to solve the time-constrained robust MDP. Extensive numerical experiments show the effectiveness of the proposed framework and algorithms.

**Strengths:**

Given that the field of robust MDP has limited large-scale experiments due to the intractability of the rectangular assumption, the idea in this paper is very nice. I agree that the rectangular assumption can be too general and lead to conservative policies, due to its inefficiency in leveraging the inherent structure of the uncertainty set. The proposed time constraint robust MDP is interesting, especially since the experiment results seem to be promising.

**Weaknesses:**

My concern is that the methods proposed in this paper might be too heuristic, given the non-stationarity and possibly non-existence of the optimal policy. As I am not very familiar with the experiment side, I am not sure if the evaluation in this paper is sufficient. See my questions for details.

**Questions:**

1. In the experiments, the oracle-TC is not always the best method. It seems that the performances are not that stable. Also, the vanilla TC outperforms in almost half cases. It is kind of anti-intuitive.

2. Compared to baseline methods, the TC-based methods show robustness to some extent, but how robust are they? What are the expected best (robust) performances?

3. In algorithm 1, what is "sample a mini-batch of transitions" for? How are the agent and adversary updated?

4. On lines 174-175, the authors claim that the rectangularity assumptions are rarely met in real-world scenarios. I think more evidence or references are required to support this claim.

**Limitations:**

yes

---

> ### Author Rebuttal · Authors · 2024-08-06
>
> Thank you for your review and the questions you raised regarding our paper. We appreciate your feedback and would like to address your concerns as follows:
>
> > My concern is that the methods proposed in this paper might be too heuristic, given the non-stationarity and possibly non-existence of the optimal policy. As I am not very familiar with the experiment side, I am not sure if the evaluation in this paper is sufficient. See my questions for details.
>
> Actually, because there is always at least one stationary optimal adversary (Iyengar, 2005), the TC hypothesis preserves optimality. Hence, under classical assumptions, the proposed method is not heuristic: the optimal policy is preserved, as stated at the beginning of Section 6.
>
> > Questions:
> > In the experiments, the oracle-TC is not always the best method. It seems that the performances are not that stable. Also, the vanilla TC outperforms in almost half cases. It is kind of anti-intuitive.
>
> We took great care to avoid over-interpretation of empirical results. Quite often in robust RL, overall scores suffer from high variance. Hence, identifying clear domination between two algorithms can be tricky.
> One could conjecture that vanilla TC policies benefits from having a reduced input space (observation only, while the others use a sequence of observations or the information of $\psi$) which might prevent overfitting, while the observation itself holds enough information for a robust policy. This seems hard to verify though.
>
> > Compared to baseline methods, the TC-based methods show robustness to some extent, but how robust are they? What are the expected best (robust) performances?
>
> The optimal best robust performance is unknown in most realistic, large robust RL benchmarks. Additionally, non convexity of optimization landscapes prevents from providing optimality certificates for solutions. So the best one can do is rank algorithms based on their average score, and recall the variance.
>
> > In algorithm 1, what is "sample a mini-batch of transitions" for? How are the agent and adversary updated?
>
> This mini-batch is used in the subsequent UpdatePolicy lines (those updates are classical SGD steps, using this minibatch). Specifically, the update in our experiments is that of TD3, but the SAC update could be dropped-in without any modification to the overall training loop.
>
> > On lines 174-175, the authors claim that the rectangularity assumptions are rarely met in real-world scenarios. I think more evidence or references are required to support this claim.
>
> It is quite hard to prove a negative, but the essence of the rectanglarity assumption is very un-natural in the first place: it implies a bicycle's parts can change mass from one time step to the other, or that the weather is drastically different a few meters apart along the same road. So although the rectangularity assumption makes theoretical analysis possible, it is commonly admitted in robust MDPs that it makes little sense in practice.

---

> > ### Comment · Reviewer_aiQz · 2024-08-11
> >
> > Thanks for the authors' rebuttal. It addressed most of my concerns.

---

> > > ### Author Response · Authors · 2024-08-13
> > >
> > > Thank you for your thoughtful review. We appreciate your feedback and the opportunity to clarify our work. We hope that our explanations regarding the preservation of the optimal policy and the interpretation of empirical results have helped to address your concerns. We believe these points underscore the value of our contributions. With this in mind, we humbly ask if you might reconsider the scores, given also that the rebuttal seems to have addressed most of your concerns.

---

### Official Review · Reviewer_ZKxz · 2024-07-08

**Soundness:** 3
**Presentation:** 3
**Contribution:** 2
**Rating:** 5
**Confidence:** 4

**Summary:**

This paper aims to develop a new framework of robust RL addressing the over-conservability issue of rectangularity and dynamic uncertainty set assumption. This framework allows for time-dependent, correlated and multifactorial disturbance to the dynamics. Three distinct algorithms are developed depending on the available information to the policy. Extensive numerical experiments are further provided to demonstrate the performance of the proposed algorithms.

**Strengths:**

- The proposed framework addresses the major drawbacks of existing robust RL approach: overly conservative due to the use of dynamic and rectangular uncerainty sets.
- Extensive experiments are provided to demonstrate the performance of the three algorithms
- The approaches can be used with existing robust value iteration approaches.

**Weaknesses:**

- Thm 2.1 only applies to the case where the policy has the exact information of \psi, which in practice is usually unavailable.
- The usefulness of this framework remains questionable, as it poses a significant challenge of constructing such complex uncertainty set with time and state-action correlation.
- The training algorithm has the same flavor of the adversarial training as those in the literature, e.g., RARL. The novelty seems rather limited as this is mostly an  experimental paper.

**Questions:**

See weaknesses above.

---

> ### Author Rebuttal · Authors · 2024-08-06
>
> We thank the reviewer for this feedback.
> We are quite surprised by the mismatch between the expressed comments and the overall grade attributed to the paper. To us, the three comments somehow discard important parts of the paper and we warmly welcome further discussion to address them if necessary.
>
> > Thm 2.1 only applies to the case where the policy has the exact information of \psi, which in practice is usually unavailable.
>
> Indeed. We don't think this is really a major limitation: there is no guarantee that an observation-based optimal policy exists for POMDPs and yet people train regular RL algorithms on real-life environments with partial observability all the time. It is quite the same thing here. What theorem 2.1 states is a general property of the Bellman operator, which induces the existence of an optimal value function when (as you accurately point out) $\psi$ is observable. This does not mean that our algorithms are only applicable in this setting (see answer below).
>
> > The usefulness of this framework remains questionable, as it poses a significant challenge of constructing such complex uncertainty set with time and state-action correlation.
>
> We beg to differ. When one drives on the highway, the transition models in different geographical conditions are strongly correlated together (by the global weather for instance) and often follow a time-constrained evolution. Similarly, a bicycle's dynamics in different states and actions are coupled by general parameters (friction coefficients for instance) that couple them together. Such uncertainty sets are very easy to construct when designing simulators (one only needs to let the global parameters vary).
> We believe there may be a misunderstanding here and warmly welcome further discussion with the reviewer to better address this concern, and understand why it seems so undue to us.
>
> > The training algorithm has the same flavor of the adversarial training as those in the literature, e.g., RARL. The novelty seems rather limited as this is mostly an experimental paper.
>
> Indeed, most algorithms in the literature (most algorithms that scale to large domains at least) are RVI-inspired (like RARL or M2TD3).  We believe this is actually a strength of the current proposal: it can be used in any previous RVI algorithm to make it better. Rejecting the paper on this basis is not a good signal for RL research: if only new algorithms were to be published, a large number of generic, important works would never receive attention.
> Additionally, we prove several properties of the TC framework, including that the optimal value function is not excluded by the TC assumption, hence we feel calling this work mostly experimental somehow discards parts of the contribution we deem important. Here again, we welcome further discussion to clarify things.

---

> > ### Comment · Reviewer_ZKxz · 2024-08-11
> >
> > The reviewer thank the authors for the feedback. They addressed my concerns. I raised the score to 5.

---

### Official Review · Reviewer_1Xhb · 2024-07-11

**Soundness:** 3
**Presentation:** 3
**Contribution:** 3
**Rating:** 6
**Confidence:** 2

**Summary:**

The paper introduces Time-Constrained Robust MDPs (TC-RMDPs) as a novel formulation to address the overly conservative nature of traditional robust RL under sa-rectangularity assumptions. The authors propose three algorithms to handle time-dependent and correlated disturbances in different situations. Extensive evaluations on continuous control benchmarks show that these algorithms outperform traditional robust RL methods in terms of balancing performance and robustness.

**Strengths:**

1. The introduction of TC-RMDPs addresses a significant limitation of overly pessimistic in traditional robust RL, making the approach more applicable to real-world scenarios with time-dependent disturbances, which is interesting and novel. In addition, the paper presents three distinct algorithms with varying levels of information usage, expanding its usage.

2. The authors provide formal theoretical guarantees for the proposed algorithms, which is appreciated.

3. Extensive experiments on continuous control benchmarks demonstrate the efficacy of the proposed methods.

**Weaknesses:**

# Major
1. The complexity of the proposed algorithms, especially Oracle-TC, may limit their practical applicability in real-world scenarios where complete environmental information is unavailable.

2. How to select the Lipschitz constant $L$ in practical scenarios? Some ablations studies and analysis regarding $L$ would be beneficial.

3. It would be better if the authors could provide some visualizations of the difference between the traditional independent transitions with the time-constrained parametric transitions.

4. What is the number of worst-case episodes in the experiments? If it is 1, maybe it would be better to display the results as the average of the worst 10% of episodes for better robustness. This is because having only 1 episode might introduce stochasticity.

# Minor
1. It is suggested to show the equation number for better illustration and reference.
2. There are some typos in the paper. For instance, in line 102, $B=\mathcal{B}(0_{\Psi},L)$ should be corrected to $B=\mathcal{B}({\Psi}_0,L)$,  the right-most column in Table 1 should be Avg. The authors should carefully review the paper for such errors.

**Questions:**

please see above

**Limitations:**

The framework assumes that the transition parameter vector $\psi$ is known during training. However, in real-world applications, it could be challenging. Also, the assumption on the uncertainty set $\Psi$ is also not practical in practice.

---

> ### Author Rebuttal · Authors · 2024-08-06
>
> We thank the reviewer for these insightful comments.
> We believe the major concerns raised are actually minor, in the sense that they can all be answered by elements already in the paper, and better explanations and phrasing. We explain why below and welcome further discussion with the reviewer.
>
> > 1. The complexity of the proposed algorithms, especially Oracle-TC, may limit their practical applicability in real-world scenarios where complete environmental information is unavailable.
>
> The rationale of proposing 3 variations on the same theme (TC, stacked-TC and Oracle-TC) is precisely designed to address your concern. As discussed in the paper, optimality can only be sought for Oracle-TC, since TC only solves a partially observable problem. Conversely, TC is designed to account for the real-world scenarios you mention. Stacked-TC is intended as an in-between solution that retains partial observability, yet recovers optimality when the underlying state can be inferred from the latest observations and actions.
> We hope this clarifies things. In our opinion, and unless we misunderstood your concern, we believe this identified weakness is already addressed in the paper.
>
> > 2. How to select the Lipschitz constant $L$ in practical scenarios? Some ablations studies and analysis regarding $L$ would be beneficial.
>
> Indeed, in practical scenarios, it is unrealistic to assume $L$ will be known. We claim TC approaches are actually quite insensitive to $L$'s value. To demonstrate this empirically, we trained our algorithms on the fixed $L=0.001$ setting and then evaluated them on environments with varying $L$ values, with order of magnitude $L=0.1$. This is already reported at line 265. We can stress this out more in the final version, with a particular emphasis on the fact that one needs not know $L$ beforehand and can make a conservative assumption on it, eventually making it a non-critical hyper-parameter.
>
> > 3. It would be better if the authors could provide some visualizations of the difference between the traditional independent transitions with the time-constrained parametric transitions.
>
> This is a very interesting question. Indeed, in the rectangular case, it is known that the minimum over the simplex of $sa$-local transition probabilities at each time step, is found on the border of this simplex. What the TC hypothesis does, is actually constrain this simplex' radius. So, in environments that respect the rectangularity assumption, we could expect that the difference you are referring to is actually the difference in diameters. But there are two key limitations to this reasoning.
> First, real-world scenarios are precisely not rectangular, and the worst cases need not be found on the border of the uncertainty set.
> But (secondly) more importantly, as stated at the beginning of Section 6, there always exists a worst-case adversarial policy (ie. a worst-case adversarial transition model) that is stationary (Iyengar, 2005). Therefore, such a transition kernel respects the TC property and can be found by our algorithms. If the worst-case dynamic transition model is unique, then it is necessarily stationary, and hence both classic RVI algorithms and TC ones should converge to it. In this case, the difference between models should be zero. But this might be unverifiable in practice because a span of dynamic models can actually lead to the optimal robust value function. So the difference between the time-constrained dynamic models and the non-time-constrained ones is eventually not very informative.
> For this reason, although your remark is very relevant, we believe visualizing such a difference would not be very informative and might be misleading. We propose to add this short discussion to the paper.
>
>
> > 4. What is the number of worst-case episodes in the experiments? If it is 1, maybe it would be better to display the results as the average of the worst 10% of episodes for better robustness. This is because having only 1 episode might introduce stochasticity.
>
> We suppose you are referring to the static evaluation case (for the dynamic evaluation case, there is a single adverasary's policy found by the algorithm).
> We checked the average score across the 10% worst transition models in that case. Although we agree with the statistical robustness argument, we would like to point out that this is eventually a different criterion overall, closer to a CVaR in spirit (or to a DR evaluation on a limited set of transition kernels). The results did not change significantly. We will add them to the appendix.

---

> > ### Comment · Reviewer_1Xhb · 2024-08-09
> >
> > Thanks for the authors' response. It addressed most of my concerns, and I believe the paper meets the acceptance threshold.

---

### Official Review · Reviewer_stMZ · 2024-07-16

**Soundness:** 3
**Presentation:** 3
**Contribution:** 3
**Rating:** 7
**Confidence:** 3

**Summary:**

The paper defines a robust training method for MDPs under uncertainty in the environment dynamics. Current methods assume sa-rectangularity, where the transition dynamics from consecutive states are independent of one another. The authors argue that this assumption is unrealistic and leads to overly conservative policies. Alternatively, this paper breaks the independence assumption in the environment dynamics to use a time-step evolution approach. This paper presents a Time-Constrained TC-MDP where the Time-Constrained Parameterized MDP kernels evolve each time-step constrained to be Lipschitz continuous. The model is trained in an adversarial manner, where the adversarial policy is learned to evolve the model dynamics but is limited to be L-close to the previous dynamics by definition. This learning method is tested under different variant conditions (cosine, exponential, linear, and logarithmic) with positive results. Robustly trained agents that learn under the worst TC adversities outperform vanilla robust algorithms that assume time independence in the model dynamics.

**Strengths:**

- The evaluation is fair. The model trained under worst-case time-constrained adversarial dynamics is then evaluated under different fixed evolving dynamics (cosine, exponential, linear, and logarithmic) that range in stochastic changes much larger than the used in learning (L=0.1 in evaluation vs L=0.001 in training).
- The definition of TC-RMDP is smartly selected to preserve stationarity, defining the state as the tuple ($S \times \Psi$)
- The TC strategy can be straightforwardly implemented in previous algorithms by limiting the search space of the adversarial $psi$ at every time-step.

**Weaknesses:**

- Reporting that the worst-case TC variants outperform vanilla adversarial models as M2TD3 or RARL is expected. As the decisions of the adversary are rather limited when compared to the vanilla algorithms. Perhaps highlighting the evaluation results (fixed or static setting) in the conclusions would help in emphasizing this.

**Questions:**

- In Algorithm 1, should is be: $b_{t+1} = \bar{\pi}(s_t, a_t, \psi_t)$ ?
- Typo in Page 7 Line 276: "Appendix G and G"

**Limitations:**

As the authors mention, this method assumes that the environment dynamics can be correctly parametrized in a confident set $Psi$.

Additionally, selecting the radius L of the proximal adversarial set of actions is not trivial. Fortunately, the authors tested the environment with a large enough L (100 times larger than the one used during training) with positive results, which validates their claim.

---

> ### Author Rebuttal · Authors · 2024-08-06
>
> We thank the reviewer for their feedback, their careful analysis of our contributions, and positive assessment of them.
>
> > Reporting that the worst-case TC variants outperform vanilla adversarial models as M2TD3 or RARL is expected. As the decisions of the adversary are rather limited when compared to the vanilla algorithms. Perhaps highlighting the evaluation results (fixed or static setting) in the conclusions would help in emphasizing this.
>
> Actually, even though it seems counter-intuitive, there is not immediate theoretical reason for TC algorithms to outperform vanilla ones, despite the fact that their action space is smaller. The reason appears quite clearly when one considers that among the optimal adversaries, at least one is stationary. This adversary is reachable by both vanilla algorithms and TC ones. Consequently the key lesson here is that vanilla methods don't find the optimal adversary in practice, and that using the TC formulation preserves optimality while helping the optimization process (as demonstrated by the empirical results). We propose to better emphasize this in the conclusions as you suggest (but with the slight nuance that dominance of TC over vanilla should not be expected in the first place).
>
> > In Algorithm 1, should is be: $b_{t+1}= \bar{\pi}(s_t, a_t, \psi_t)$
>
> Both notations seem acceptable, since $\psi_t$ is within the arguments of $\bar{\pi}$. In other words, since $\psi_{t+1}=\psi_t+b_t$, the two notations are equivalent. We chose to drop the $b_t$ notation for readability.

---

> > ### Comment · Reviewer_stMZ · 2024-08-11
> >
> > Thanks to the authors for the clarification, It addresses my concerns.

---

### Decision · Program_Chairs · 2024-09-25

**Decision:**

Accept (poster)

**Comment:**

Most existing works on robust MDP make the (s,a) rectangularity assumption, which inherently assumes that the transition dynamics from a given current state to the next states are independent even for current states closer to each other. This assumption is strong, and may not be applicable in real-world problems. This paper proposes a new time-constrained robust MDP (TC-RMDP) formulation that considers correlated and time-dependent disturbances. The paper proposes three algorithms to solve this problem, each using varying levels of environmental information. The paper provides an extensive simulation-based evaluation of the proposed algorithms using standard benchmarks.


We received four expert reviews, with the scores, 7, 6, 5, 5, and the average score is 5.75. Reviewers gave positive comments about the novelty of presenting the TC-RMDP, the novelty of three algorithms, and the extensive simulation-based evaluations of the algorithms. The reviewers are also happy about the quality of the presentation.

The reviewers have also provided some very useful comments to improve the paper. Reviewer 1Xhb has asked about the complexity of proposed algorithms, selecting the parameter L, and providing a visualization/figure of the traditional (s,a) rectangular uncertainty set and uncertainty considered in this paper. Reviewer ZKxz has asked to include some specific examples of the practical applications where such uncertainty sets are relevant. I think all these are important points, and I recommend authors to include these details in their final manuscript. .